



# Global datasets of hourly carbon and water fluxes simulated using a satellite-based process model with dynamic parameterizations

Jiye Leng[1], Jing M. Chen[1], Wenyu Li[1], Xiangzhong Luo[2], Mingzhu Xu[3], Jane Liu[1], Rong Wang[3], Cheryl Rogers[4], Bolun Li[5], Yulin Yan[3]

[1] Department of Geography and Planning, University of Toronto, Toronto, ON M5S 3G3, Canada.
[2] Department of Geography, National University of Singapore, 1 Arts Link, 117570 Singapore.
[3] School of Geographical Science, Fujian Normal University, Fuzhou, 350007, China.
[4] School of Earth, Environment and Society, McMaster University, 1280 Main Street West, Hamilton, ON L8S 4K1, Canada.
[5] School of Geographical Sciences, Nanjing University of Information Science and Technology, Nanjing, 210044, China.

*Correspondence to*: Jing M. Chen (jing.chen@utoronto.ca)

**Abstract.** Diagnostic terrestrial biosphere models (TBMs) forced by remote sensing observations have been a principal tool to provide benchmarks on global gross primary productivity (GPP) and evapotranspiration (ET). However, these models often estimate GPP and ET at coarse daily or monthly step, hindering analysis of ecosystems dynamics at the diurnal (hourly) scales, and prescribe some essential parameters (i.e., the Ball-Berry slope ($m$) and the maximum carboxylation rate at 25 °C ($V_{cmax}^{25}$)) as constant, inducing uncertainties in the estimates of GPP and ET. In this study, we present hourly estimation of global GPP and ET datasets at a 0.25° resolution from 2001 to 2020 simulated with a widely used diagnostic TBM – Biosphere-atmosphere Exchange Process Simulator (BEPS). We employed eddy covariance observations and machine learning approaches to derive and upscale the seasonally varied $m$ and $V_{cmax}^{25}$ for carbon and water fluxes. The estimated hourly GPP and ET are validated against flux observation, remote sensing, and machine learning-based estimates across multiple spatial and temporal scales. The correlation coefficients ($R^2$) and slopes between hourly tower-measured and modeled fluxes are: $R^2 = 0.83$, regression slope = 0.92 for GPP and, $R^2 = 0.72$, regression slope = 1.04 for ET. At the global scale, we estimated a global mean GPP of $137.78 \pm 3.22$ Pg C yr$^{-1}$ (mean $\pm$ 1 SD) with a positive trend of 0.53 Pg C yr$^{-2}$ (p < 0.001), and ET of $89.03 \pm 0.82 \times 10^3$ km$^3$ yr$^{-1}$ with a slight positive trend of $0.10 \times 10^3$ km$^3$ yr$^{-2}$ (p < 0.001) from 2001 to 2020. The spatial pattern of our estimates agrees well with other products, with $R^2 = 0.77 - 0.85$ and $R^2 = 0.74 - 0.90$ for GPP and ET, respectively. Overall, this new global hourly dataset serves as a 'handshake' among process-based models, remote sensing, and the eddy covariance flux network, providing a reliable long-term estimate of global GPP and ET with diurnal patterns and facilitating studies related to ecosystem functional properties, global carbon, and water cycles. The hourly and accumulated daily GPP and ET estimates are available at https://doi.org/10.5281/zenodo.8240492 (Leng et al., 2023).

## 1 Introduction

Terrestrial photosynthesis (gross primary productivity, GPP) and evapotranspiration (ET) play pivotal roles in the intricate dynamics of the global carbon and hydrological cycles (Piao et al., 2020; Liu et al., 2003; Jasechko et al., 2013). Enhancing our understanding of the exchanges of carbon and water between terrestrial ecosystems and the atmosphere holds paramount importance for understanding and monitoring the Earth system (Ryu et al., 2019; Chen et al., 2019; Jung et al., 2010). However, large discrepancies exist in the estimation of global GPP and ET fluxes by various models varied from 92.7 to 168.7 Pg C yr$^{-1}$ (Zheng et al., 2020) and from 63.3 to $105.4 \times 10^3$ km$^3$ yr$^{-1}$ (Li et al., 2023; Yu et al., 2022), respectively. The present uncertainties predominantly arise from multiple constraining factors, encompassing, although not exclusively, oversimplified model structure (Yuan et al., 2010; Liang et al., 2013; Wang et al., 2020; Tagesson et al., 2021; Moreno et al., 2012; Luo et al., 2018), in adequate representation of plant functional traits (Chen et al., 2022; Chen et al., 2013; Bonan et al., 2011; Wilson et al., 2000; Wang et al., 2007; Franks et al., 2018), coarse resolution of climate forcing, and also inconsistency of global land



cover maps (Bonan, 2019; Gamon et al., 2004; Bonan et al., 2002). As a consequence, the precise quantification of global carbon and water fluxes remains an enduring challenge.

State-of-the-art terrestrial biosphere process-based models (TBMs), coupling the biogeochemical and biogeophysical
processes in the soil-vegetation-atmosphere continuum, have been developed to estimate global carbon and water fluxes (Cao and Woodward, 1998; Sitch et al., 2003). Differing from the prognostic TBMs (i.e., TRENDY), diagnostic TBMs offer a heightened level of reliability as benchmarks for GPP and ET, because of their alignment with remote sensing-derived appraisals of plant structural conditions (Liu et al., 1997; Chen et al., 1999; Chen et al., 2012; Luo et al., 2018; Liu et al., 2003). The diagnostic TBMs often adopt the scheme by integrating an enzyme-kinetic biochemical photosynthesis model by Farquhar
et al. (1980) with the Ball-Woodrow-Berry (BWB) stomatal conductance model (Ball et al., 1987). The maximum carboxylation rate ($V_{cmax}$) quantifies the leaf photosynthetic capacity, and its normalized form at 25 °C ($V_{cmax}^{25}$) is an essential parameter used to estimate carbon fluxes in TBMs. Besides, the Ball-Berry slope, $m$, serves as the parameter that balances the rate of carbon gain and water loss of plants by controlling the modeled stomatal conductance in simulating the photosynthetic process. In regional and global ecosystem modeling, current TBMs tend to assign $V_{cmax}^{25}$ as a fixed parameter varied by plant
functional types (PFTs), which were typically estimated from a measurement-based database (Kattge et al., 2009), and to assign $m$ as a constant. However, in recent studies, $V_{cmax}^{25}$ and $m$ have been found to vary across PFTs (Chen et al., 2022; Smith et al., 2019; Lin et al., 2015; Bauerle et al., 2014; Miner et al., 2017) and seasonally (Miner and Bauerle, 2017; Misson et al., 2004; Wolz et al., 2017; Luo et al., 2021; Croft et al., 2017; Liu et al., 2023; Leng et al., under review). Therefore, incorporating prescribed constant $V_{cmax}^{25}$ and $m$ in TBMs may induce uncertainties in modeling global GPP and ET (Ryu et al.,
2019; Miner and Bauerle, 2017).

Process-based models necessitate the inclusion of parameters, yet several of those parameters are challenging to determine through empirical data alone. Machine learning techniques can be employed to acquire parameterizations that effectively depict the observed ground truth. This results in a model that combines the benefits of physical modeling, leveraging its theoretical foundations, with the adaptive capabilities of machine learning techniques. These data-driven adjustments and optimizations
enhance the modeling of spatiotemporal patterns in carbon and water cycles (Reichstein et al., 2019). Recent studies have utilized models with data-driven parameterizations to estimate global GPP and ET (Zhao et al., 2019; Koppa et al., 2022; Hu et al., 2021; Ma et al., 2022). Employing data-driven parameterization not only improve the estimation of carbon and water fluxes, but also enables a deeper understanding of the dynamics and mechanisms governing ecosystem functions.

Refining GPP and ET estimations from daily to hourly scales provides valuable insights into the diurnal patterns of plant-
atmosphere interactions, particularly in relation to extreme climate events (Hashimoto et al., 2020; Bodesheim et al., 2018; Duarte Rocha et al., 2022). Air temperature and vapor pressure deficit are higher in the morning than in the afternoon (Goulden et al., 2004; Lin et al., 2019), and leaf water potential decreases from morning to afternoon due to the water loss from transpiration (Neumann and Cardon, 2012; Lee et al., 2005). These processes can lead to different responses of photosynthesis and transpiration to environment, which only the datasets with diurnal variations can track (Zhang et al., 2023b). Besides, on
the diurnal scale, the temporal patterns of GPP and ET are influenced mostly by radiation and the structural characteristics of plant canopies, specifically by the variable contributions of sunlit and shaded leaves (Chen et al., 1999; Ryu et al., 2011; Nelson et al., 2020; Luo et al., 2018). Sunlit leaves simultaneously absorb both direct and diffuse radiation, making their photosynthesis primarily limited by Rubisco. In contrast, shaded leaves solely absorb diffuse radiation, making their photosynthesis constrained by incoming solar energy (Ju et al., 2006; Wang and Leuning, 1998; Urban et al., 2007; Luo et al.,
2018). Therefore, differentiating hourly water and carbon fluxes into sunlit and shaded proportions would further untangle the complex interactions between plant physiological and canopy structural components, and the ambient environment.

Here we provide the first dataset of global hourly carbon and water fluxes using a two-leaf satellite-based TBM with dynamic parameterizations. We developed the model by 1) inversing seasonally variable $m$ and $V_{cmax}^{25}$ from measured eddy covariance data, 2) upscaling derived $m$ and $V_{cmax}^{25}$ to modelling grids with machine learning algorithms, and 3) estimating hourly carbon

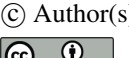



and water fluxes in a two-leaf hourly process-based model with seasonally-spatially variable $m$ and $V_{cmax}^{25}$. We evaluated the effectiveness of the dataset in capturing the spatial, temporal, and interannual patterns in GPP and ET from the eddy covariance sites. We also intercompared the dataset with other global GPP and ET datasets to assess their spatial and interannual variations.

**2 Data and Methodology**

**2.1 Data from the eddy covariance towers**

The FLUXNET2015 (http://www.fluxdata.org) dataset includes the measured and postprocessed carbon fluxes, energy fluxes, and meteorological variables over more than 200 sites around the globe in a standard format (Pastorello et al., 2020). In this study, we selected 136 sites (809 site-years) (Table S1) based on the availability and quality of measured fluxes and meteorological conditions (the quality control flag for fluxes smaller than 2 and data gap less than 20% for a site-year). For

the independent validation, we randomly selected 20% of the total sites to evaluate the accuracy and applicability of the process model with dynamic parameterizations.

Half-hourly meteorological records in the FLUXNET2015 dataset were aggregated into hourly records and used to drive BEPS. Gap-filled incoming shortwave radiation (SW_IN_F), air temperature (TA_F), vapor pressure deficit (VPD_F), precipitation (P_F), and wind speed (WS_F) were selected as the forcing meteorological variables to derive monthly variable $m$ and $V_{cmax}^{25}$

at each site. We chose to use GPP partitioned from Net Ecosystem Exchange (NEE) based on a night-time method with a variable friction velocity (u*) threshold (GPP_NT_VUT) and the gap-filled latent heat flux (LE_F_MDS) for each site year as the targeted fluxes for $m$ and $V_{cmax}^{25}$ estimation. In the validation, we only chose to use GPP and latent heat flux with quality control flags smaller than 2 to compare with the modeled GPP and ET at the hourly, daily, and annual scales.

**2.2 Gridded data for the globe**

The global datasets used in this study are shown in **Table 1**. The meteorological variables were collected from the fifth generation European Center for Medium-range Weather Forecasts (ECMWF) reanalysis (ERA-5) with hourly records and a spatial resolution of 0.25°. In this study, we obtained the hourly incoming shortwave radiation (SW, W m$^{-2}$), air temperature at 2 m ($T_a$, °C), dew point temperature ($T_d$, °C), precipitation (mm h$^{-1}$), 10 m u- and v-component of wind ($u_{10}$ and $v_{10}$, m s$^{-1}$), soil water content (SWC, m$^3$ m$^{-3}$), and snow depth (SWD, m). The relative humidity (RH) was calculated from air temperature

and dew point temperature as:

$$RH = e^{\frac{17.269T_d}{T_d+237.2} - \frac{17.269T_a}{T_a+237.2}} * 100\% \quad (1)$$

The wind speed (WS) was calculated from the horizontal (u) and the vertical (v) component of wind measured at a height of 10 m as:

$$WS = \sqrt{u_{10}^2 + v_{10}^2} \quad (2)$$

The 8-day GLOBMAP leaf area index (LAI) dataset at 8 km resolution was used to drive the BEPS model. The LAI dataset was produced by quantitative fusion of Moderate Resolution Imaging Spectroradiometer (MODIS) and historical Advanced Very High Resolution Radiometer (AVHRR) data (Liu et al., 2012). In this study, we smoothed and interpolated the 8-day LAI data into continuous daily LAI series using the Locally Adjusted Cubic-spline Capping (LACC) algorithm (Chen et al., 2006). To account for nonrandomness of the leaf distribution within a canopy (Chen et al., 1997), we collected the global

clumping index map retrieved from the MODIS Bidirectional Reflectance Distribution Function (BRDF) products (He et al., 2012) at a spatial resolution of 500 m.

We also used a leaf chlorophyll content (LCC) dataset to represent leaf physiological status from 2001 to 2020. It was derived based on MODIS data by coupling a leaf optical properties model and a canopy bidirectional reflectance model (Xu et al., 2022). We also used the LACC algorithm to interpolate the 8-day LCC into daily LCC series.

In addition, we obtained the yearly global land cover map with the IGBP classification scheme from the MCD12C1 product (Friedl and Sulla-Menashe, 2015). We collected the soil texture, fraction of clay, soil, and sand from the Harmonized World Soil Database (HWSD) v1.2 to parameterize the soil properties in BEPS. We also acquired the monthly $CO_2$ concentration measurements from the NOAA Earth System Research Laboratory (ESRL). We resampled the LAI, CI, LCC, soil properties, and land cover maps into a spatial resolution of 0.25°.


**Table 1 Global datasets used in the BEPS model with dynamic parameterizations.**

| Variable | Datasets/Source |
| --- | --- |
| Incoming shortwave radiation | ERA-5<br>https://cds.climate.copernicus.eu/cdsapp#!/dataset/reanalysis-era5-single-levels |
| Air temperature | ERA-5 |
| Dew point temperature | ERA-5 |
| Precipitation | ERA-5 |
| Wind speed | ERA-5 |
| Soil water content | ERA-5 |
| Snow depth | ERA-5 |
| LAI | GLOBMAP (Liu et al., 2012) |
| Clumping index | He et al. (2012) |
| Leaf chlorophyll content | Xu et al. (2022) |
| Land cover map | MCD12C1<br>https://lpdaac.usgs.gov/products/mcd12c1v006/ |
| Soil texture map | Harmonized World Soil Database v1.2<br>https://www.fao.org/land-water/databases-and-software/hwsd/en/ |
| $CO_2$ concentration | NOAA's Earth System Research Laboratories<br>https://gml.noaa.gov/ccgg/trends/ |

**2.3 The process-based model with dynamic parameterizations**

**2.3.1 Parameter optimization for BEPS**

The schematic overview of the methodology and data sources is shown in **Figure 1**. The TBM used in this study is the Biosphere-atmosphere Exchange Process Simulator (BEPS), renamed from the Boreal Ecosystem Productivity Simulator. BEPS is a two-leaf diagnostic enzyme-kinetic model and has been intensively adopted for quantifying carbon and water fluxes over various biomes and over the globe (Luo et al., 2019; Liu et al., 2003; Chen et al., 1999; Chen et al., 2019; Chen et al., 2012). The newly revised BEPS v4.10 adopts hourly meteorological variables (i.e., incoming shortwave radiation, air

temperature, vapor pressure deficit, precipitation, and wind speed) to model hourly carbon and water fluxes. The shortwave radiation and leaf temperature are separately calculated for sunlit and shaded leaf groups (Chen et al., 1999). Leaf-level photosynthetic rate and stomatal conductance are obtained through the coupling of the Farquhar scheme (Farquhar et al., 1980) and Ball-Woodrow-Berry stomatal conductance model (Ball et al., 1987) using a cubic analytical solution (Baldocchi, 1994).



The leaf-level transpiration is then obtained based on the Penman-Monteith equation. The canopy-level GPP and ET are calculated as:

$$GPP = GPP_{sunlit} * LAI_{sunlit} + GPP_{shaded} * LAI_{shaded} \tag{3}$$

$$ET = E + T_{sunlit} * LAI_{sunlit} + T_{shaded} * LAI_{shaded} \tag{4}$$

where $LAI_{sunlit}$ and $LAI_{shaded}$ are the LAI for sunlit and shaded leaf groups, respectively. A detailed description of the main modules in BEPS is provided in the supplement.

We recently developed a new parameter optimization algorithm (Leng et al., under review) for BEPS, using measured GPP and ET fluxes to constrain the simulations from BEPS through optimizing key photosynthesis and stomatal conductance model parameters (i.e., $V_{cmax}^{25}$ and $m$). The Bayesian parameter optimization with the carbon-water coupling cost function (Eqn. S14) has been validated to efficiently and accurately estimate $m$ and $V_{cmax}^{25}$ and to improve the modeling of carbon and water fluxes. We updated $m$ and $V_{cmax}^{25}$ in each iteration to allow BEPS to model carbon and water fluxes as close as possible to the measured fluxes. The detailed description of the algorithm is provided in the supplementary materials. In this study, we adopted the parameter optimization algorithm to estimate monthly $m$ and $V_{cmax}^{25}$ for each site-year of flux sites.

**2.3.2 Dynamic parameterizations for BEPS using machine learning**

Two separate Random Forest Regressors were trained using the combinations of the plant properties (functional types, LAI, LCC) and environmental conditions (meteorological variables, soil types, soil water content) with the optimized $m$ and $V_{cmax}^{25}$, respectively. The plant functional types and soil types were encoded with the one-hot encoder. The meteorological variables and soil water content were collected according to the time and location of each $m$ and $V_{cmax}^{25}$ retrievals. 80% of data were randomly selected to train the Random Forest Regressors with five-fold cross-validation to determine the hyperparameters and 20% of data were used to evaluate the performance of the trained regressors in each round of calibration. All sites were used in the training process for upscaling $m$ and $V_{cmax}^{25}$ for the site level to gridded maps. The global monthly $m$ and $V_{cmax}^{25}$ time series were separately generated for 2001-2020 using the gridded feature data in the Random Forest Regressors. Then, BEPS adopted the monthly $m$ and $V_{cmax}^{25}$ trained from machine learning models based on the FLUXNET2015 dataset, hereafter referred to as the BEPS with dynamic parameterizations (BEPS-DP).

BEPS-DP used the meteorological variables from ERA-5 as climate forcing and the LAI from GLOBMAP to model the GPP and ET for sunlit and shaded leaf groups at an hourly step. The simulations of photosynthetic rate and evapotranspiration were iterated ten times for each hour to obtain the final hourly estimate. Then the hourly estimates of GPP were aggregated to daily and annual time steps for model validations and evaluations.

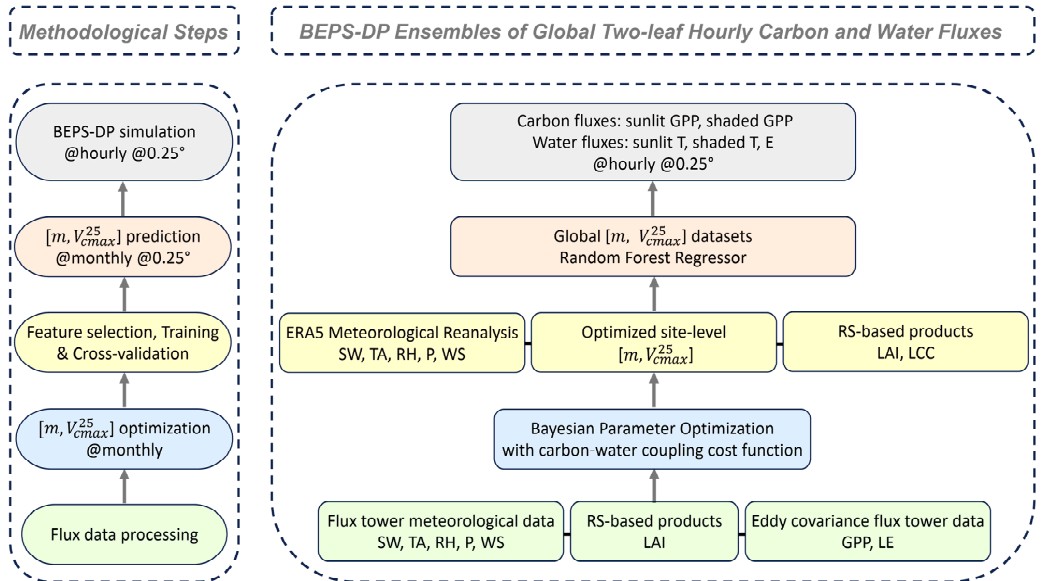

**Figure 1. Schematic overview of the methodology and data products of the BEPS model with dynamic parameterizations (BEPS-DP). The flow diagrams show the methodological steps (left) and the details (right) for the BEPS-DP datasets of global hourly two-leaf carbon and water fluxes. SW (shortwave radiation, W m$^{-2}$), TA (air temperature, °C), RH (relative humidity, %), P (precipitation, mm h$^{-1}$), WS (wind speed, m s$^{-1}$), GPP (gross primary productivity, g C m$^{-2}$ h$^{-1}$), LE (latent heat, W m$^{-2}$)**

### 2.4 Model validation and evaluation

We validated the efficacy, accuracy, and applicability of the BEPS-DP using the fluxes of 136 sites (809 site-years), as shown in Figure 2. Three metrics, the coefficient of determination (R$^2$), the root mean square error (RMSE), and the slope between the observations and simulations were adopted to evaluate the performance of BEPS-DP. In the model-training process, a five-fold cross validation method was adopted to tune the hyperparameters in the random forest regressors. After the training, the features in the independent validation set were used as input in the random forest regressors to generate monthly $m$ and $V_{cmax}^{25}$. The BEPS model was run with the predicted $m$ and $V_{cmax}^{25}$ and other driving force data at the site level to simulate hourly GPP and ET in each site year in the independent validation set. Then the simulated GPP and ET were compared with the GPP and ET estimated from the eddy covariance at hourly, daily, and annual scales.


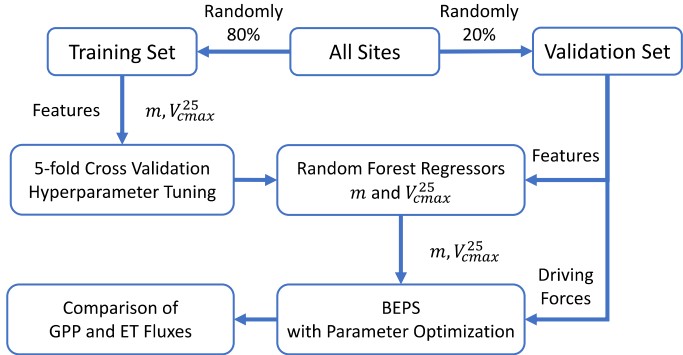

**Figure 2. Flowchart for the validation of the BEPS model with dynamic parameterizations at the site level.**

In addition, at the global level, we compared the modeled carbon and water fluxes from BEPS-DP with five other gridded flux products (one using machine learning methods, two from process-based models, one from LUE models, and one from remote sensing data), from the perspectives of global total values, gridded values, and mean annual sum patterns. The FLUXCOM ensemble datasets are widely used as the reference data in global long-term carbon and water cycles studies (Ryu et al., 2019; Tagesson et al., 2021). FLUXCOM datasets include the ensemble GPP and ET fluxes derived from three machine learning methods, at 0.5° spatial resolution and daily temporal resolution since 1979 (RS+METEO setup, i.e., remote sensing and meteorological data driven) and at 0.0833° spatial resolution and 8-daily temporal resolution since 2001 (RS setup, i.e., remote sensing driven only) (Jung et al., 2019; Jung et al., 2020). We used the RS+METEO product to intercompare the global long-term GPP and ET estimates. Process-based global GPP estimates were obtained from the Breathing Earth System Simulator (BESS) v2.0, which generates global carbon and water fluxes at 0.05° resolution at a daily time step (Ryu et al., 2011; Li et al., 2023). Process-based global ET estimates were adopted from BESS and the Global Land Evaporation Amsterdam Model (GLEAM) (Martens et al., 2017; Miralles et al., 2011). LUE-based GPP and ET estimates were collected from the Global LAnd Surface Satellite (GLASS) datasets (Liang et al., 2021). We also used the GOSIF GPP derived from remote-sensing solar induced fluorescence (SIF) to intercompare the long-term GPP trends (Li and Xiao, 2019). All comparisons were conducted at an annual time step and 0.25° resolution for the overlapped years in 2001-2020 of each flux product in this study.

## 3. Results

### 3.1 Evaluation of the BEPS model with dynamic parameterizations

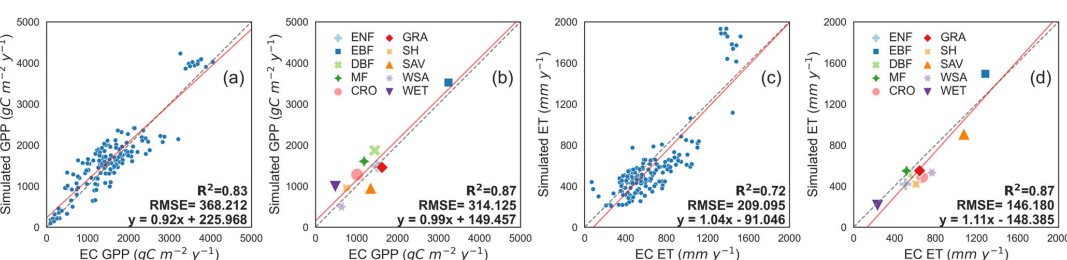

**Figure 3. Comparisons between fluxes measured at eddy covariance towers and simulated from the BEPS model with dynamic parameterizations in the independent validation set (site-year, $n$ = 152): annual summed (a) GPP and (c) ET in all the site-years; the**





**average annual summed (b) GPP and (d) ET for each biome. The red lines are the regression lines and the grey dotted lines are the 1:1 lines. The equations at the right bottom of each panel are the regression equations derived from all the site-years and from all the biomes, respectively.**

In general, BEPS-DP could effectively reproduce the temporal variations (hourly, daily, and annual) as well as the spatial differences in the tower-based GPP and ET across the majority of sites in the independent validation set (Figure 3 and Figure

4). BEPS-DP demonstrated the efficacy to explain approximately 83% and 72% of the spatial-temporal variations in GPP and ET across all validation sites and site years (Figure 3). For all the site years, the regression slopes are 0.92 and 1.04 and the root mean square errors (RMSE) are 368.212 g C m$^{-2}$ yr$^{-1}$ and 209.095 mm yr$^{-1}$ for annual scale GPP and ET comparisons, respectively. For all PFTs in the independent validation set, BEPS-DP also showed a good performance in simulating the GPP and ET at most sites (Figure 3). BEPS-DP can explain 87% of the average annual summed of GPP and ET per each biome,

with the RMSE of 314.125 g C m$^{-2}$ yr$^{-1}$ and 146.18 mm yr$^{-1}$ in GPP and ET, respectively. Although the average annual summed ET were slightly overestimated for the evergreen broadleaf forests (EBF) and the average annual summed GPP were overestimated for the wetland ecosystems in the comparisons, the regression slopes are 0.99 and 1.11 between simulated GPP, ET and the GPP, ET estimated from eddy covariance measurements.

At the hourly scale, the coefficients of determination ($R^2$) of GPP varied from 0.57 to 0.96 with 82.9% of them over 0.80 and

all of them being statistically significant ($P < 0.001$). The regression slopes between simulated GPP and GPP estimated from eddy covariance ranged from 0.43 to 1.58. The highest regression slopes were found in CRO and GRA. The $R^2$ of ET varied from 0.45 to 0.95 with 70.4% of them over 0.80. The regression slope between simulated ET and measured ET ranged from 0.44 to 1.08. At the daily scale, the $R^2$ of GPP and ET were slightly lower than the $R^2$ of hourly GPP and ET in the comparison but still with $R^2$ over 0.75 for both GPP and ET in most of the site years. However, smaller slopes were found in the

comparisons of daily fluxes due to the larger intercepts in the regression. We also compared the distribution of $R^2$ and regression slope of GPP and ET across all PFTs. At the hourly scale, the $R^2$ and regression slopes of GPP and ET were close to 1.0 in the forest ecosystems (ENF-evergreen needleleaf forest, EBF-evergreen broadleaf forest, DBF-deciduous broadleaf forests, and MF-mixed forests). But the regression slopes were low in CRO (cropland), SAV (savannas), and WSA (woody savannas), although the $R^2$ in such biomes exhibited the range of 0.6-0.8. At the daily scale, the regression slopes were much

smaller than the regression slopes of hourly fluxes because the regression intercepts were close to 0 in hourly fluxes but high in the comparisons of daily fluxes. Overall, the $R^2$ of both hourly and daily fluxes exhibited exponential distributions with peak close to 1.0 (Figure 4a, Figure 4c, Figure 4i, and Figure 4k) while the regression slopes of fluxes exhibited normal distributions (Figure 4b, Figure 4d, Figure 4j, and Figure 4l).

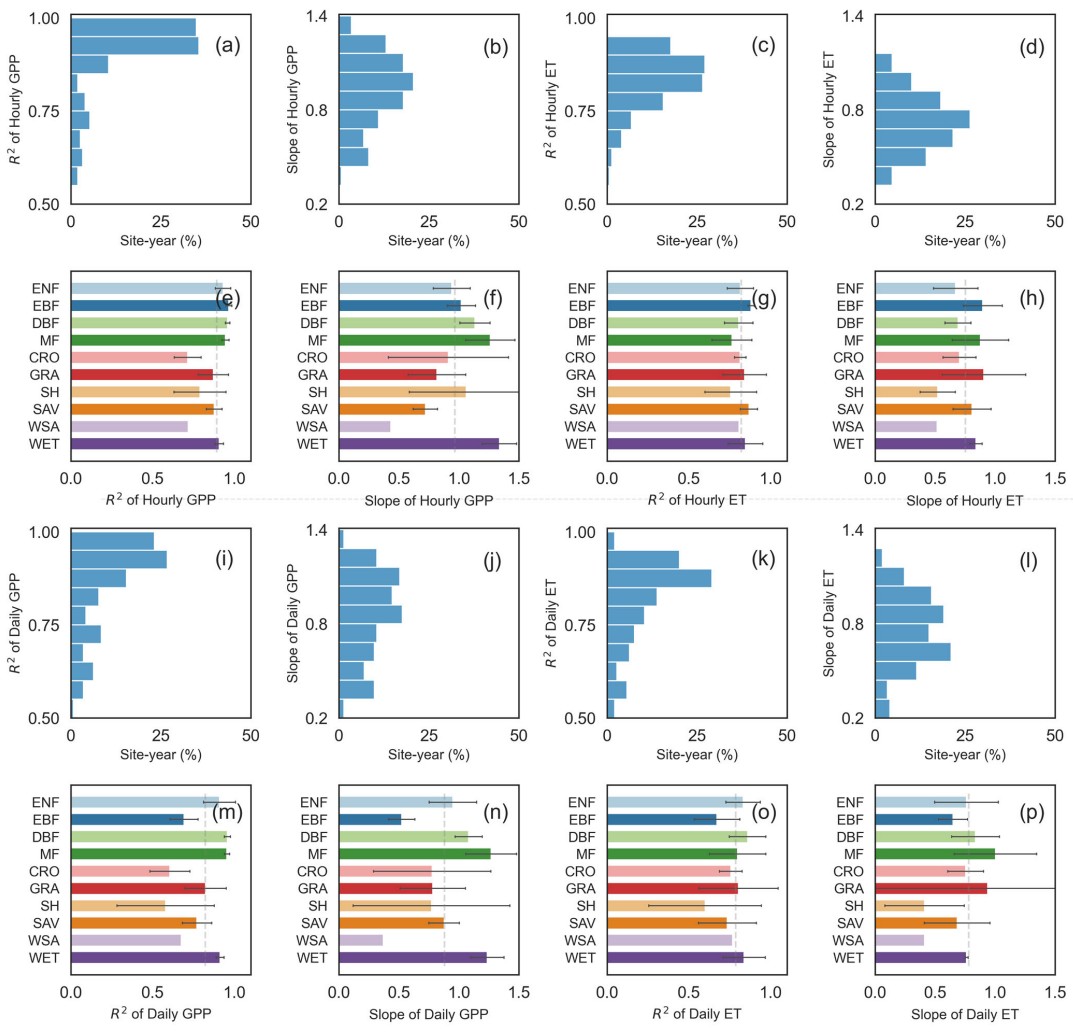


**Figure 4. Evaluation of modeled hourly and daily fluxes against the eddy covariance data in the independent validation set: site-year percentage of R² in (a) hourly GPP; (c) hourly ET; (i) daily GPP; (k) daily ET; site-year percentage of regression slopes in (b) hourly GPP; (d) hourly ET; (j) daily GPP; (l) daily ET; the mean and standard deviation (SD) of R² in each PFT in (e) hourly GPP; (g) hourly ET; (m) daily GPP; (o) daily ET; the mean and standard deviation of regression slopes in each PFT in (f) hourly GPP; (h)**
**hourly ET; (n) daily GPP; (p) daily ET. The dashed grey lines in (e) – (h) and (m) – (p) indicate the mean of R² and regression slopes for all PFTs in GPP and ET.**

## 3.2 The spatial and temporal patterns of the global carbon and water fluxes




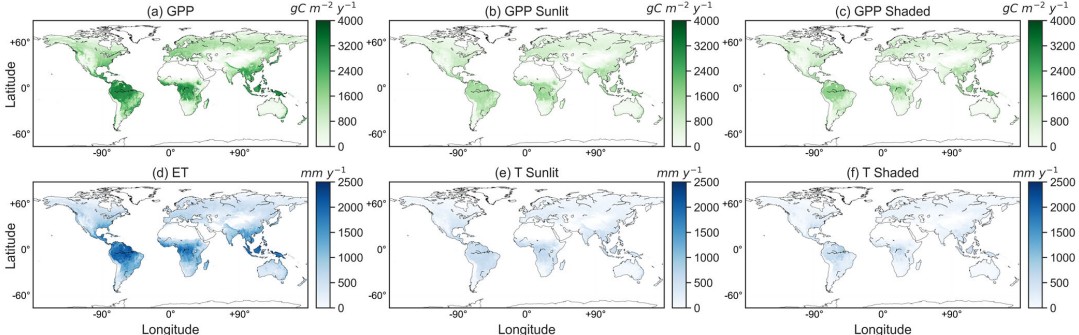

**Figure 5. Spatial patterns of global GPP and ET by the BEPS model with dynamic parameterizations during 2001-2020: (a) annual mean total GPP; (b) annual mean GPP of sunlit leaves; (c) annual mean GPP of shaded leaves; (d) annual mean total ET; (e) annual mean T of sunlit leaves; (f) annual mean T of shaded leaves.**


A global hourly GPP and ET dataset at a spatial resolution of 0.25° was generated from 2001 to 2020 using BEPS-DP with gridded driving forces (Figure 5). The 20-year averaged global GPP across the vegetated area was $137.78 \pm 3.22$ Pg C yr$^{-1}$ (Figure 5a), in which the global GPP of sunlit leaves was $73.44 \pm 1.46$ Pg C yr$^{-1}$ (Figure 5b), and the global GPP of shaded leaves was $64.34 \pm 1.79$ Pg C yr$^{-1}$ (Figure 5c). The 20-year averaged global ET across the vegetated area was $89.03 \pm 0.82 \times$

$10^3$ km$^3$ yr$^{-1}$ (Figure 5d), in which the global T of sunlit leaves was $27.72 \pm 0.23 \times 10^3$ km$^3$ yr$^{-1}$ (Figure 5e), and the global T of shaded leaves was $19.00 \pm 0.57 \times 10^3$ km$^3$ yr$^{-1}$ (Figure 5f). At the global scale, the spatial distributions of 20-year GPP and ET coupled well. Both GPP and ET were high over the tropical and subtropical areas, such as the Amazon, Central Africa, and Southeast Asia, where the water and temperature conditions allow intensive photosynthesis and transpiration. Moderate levels of GPP and ET were observed in temperate and subhumid regions, while the lowest GPP and ET values were typically found

in arid or cold regions with limited precipitation or low temperature. In addition, the global hourly GPP and ET dataset can also track the diurnal changes from dawn to dusk at UTC time (Figure 6). At UTC 6 h, high GPP and ET were observed in Southeast Asian areas while at UTC 18 h, high GPP and ET were found in American regions.

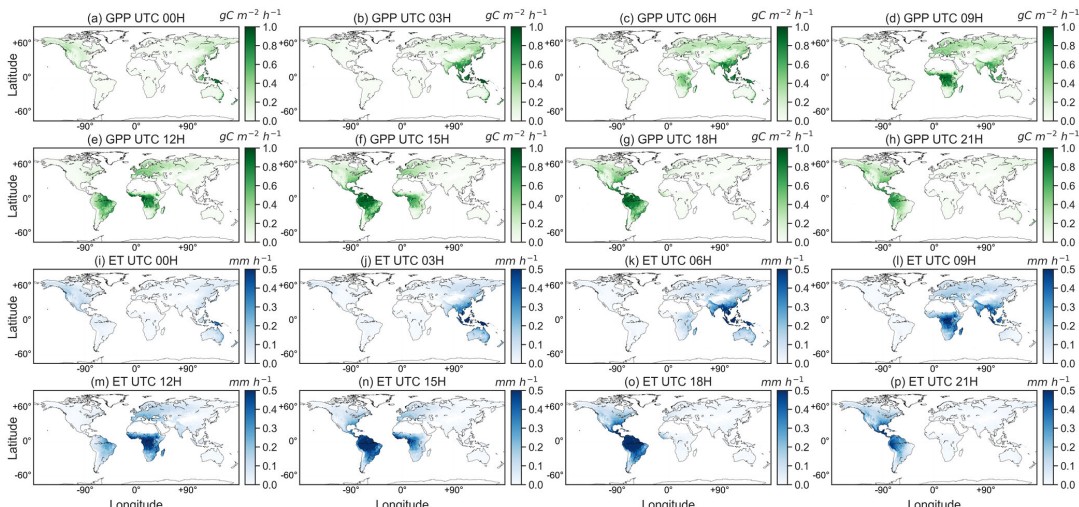

**Figure 6. Spatial diurnal patterns of global GPP and ET by the BEPS model with dynamic parameterizations during 2001-2020: (a) – (h) averaged hourly GPP at UTC time with 3 h intervals, respectively; (i) – (p) averaged hourly ET at UTC time with 3 h intervals, respectively.**

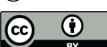
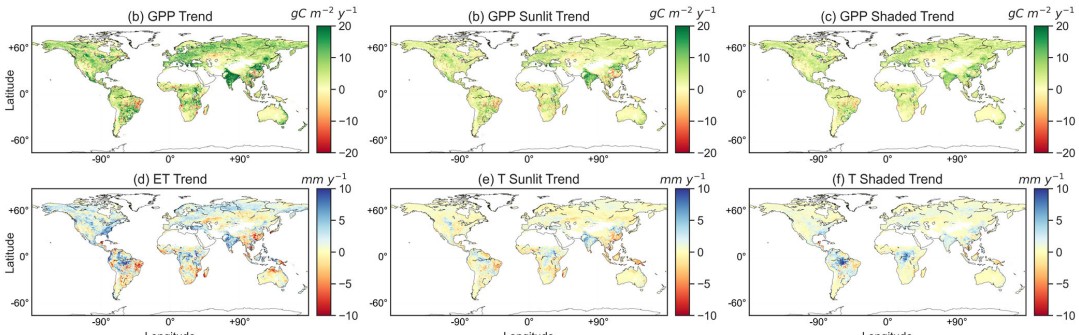

**Figure 7. Spatial patterns of global GPP and ET trends by the BEPS model with dynamic parameterizations during 2001-2020: (a) trend of annual total GPP; (b) trend of annual GPP of sunlit leaves; (c) trend of annual GPP of shaded leaves; (d) trend of annual total ET; (e) trend of annual T of sunlit leaves; (f) trend of annual T of shaded leaves.**

The long-term trends of GPP and ET from 2001 to 2020 were determined with a linear regression analysis per pixel (Figure 7). Both coupling and decoupling spatial patterns between GPP and ET were observed at the global scale. Approximately 83.3% of the vegetated area showed increased GPP trends, in which 46.8% of the vegetated area showed increased GPP trends with more than 5 g C m$^{-2}$ yr$^{-2}$ from 2001 to 2020. Approximately 70.1% of the vegetated area showed increased ET trends, and 7.8% of the vegetated area showed increased ET trends with more than 5 mm yr$^{-2}$. The decreased GPP trend was found in the tropical areas, specifically in the Amazon area. The decreased ET trend was found in the Amazon Forest, coupled with the trend of GPP, but also found in South Africa, Southwest Asia, and North Australia, showing opposite trends to GPP. More decreased patterns were observed in GPP and T of sunlit leaves (Figure 7a and Figure 7d) than shaded leaves (Figure 7c and Figure 7f), contributing the decreased trends of total GPP and ET. By incorporating measured fluxes in model parameterizations, BEPS-DP combines the benefits of ground measurements and the sophisticated structure of process-based models. We believe that the integration in this study allows for combining the strengths of both approaches, resulting in improved accuracy in estimating carbon and water fluxes at the global scale and finer temporal resolution base on the point-to-point comparisons (Figure 3 and Figure 4).

### 3.3 Comparisons with other global GPP and ET products



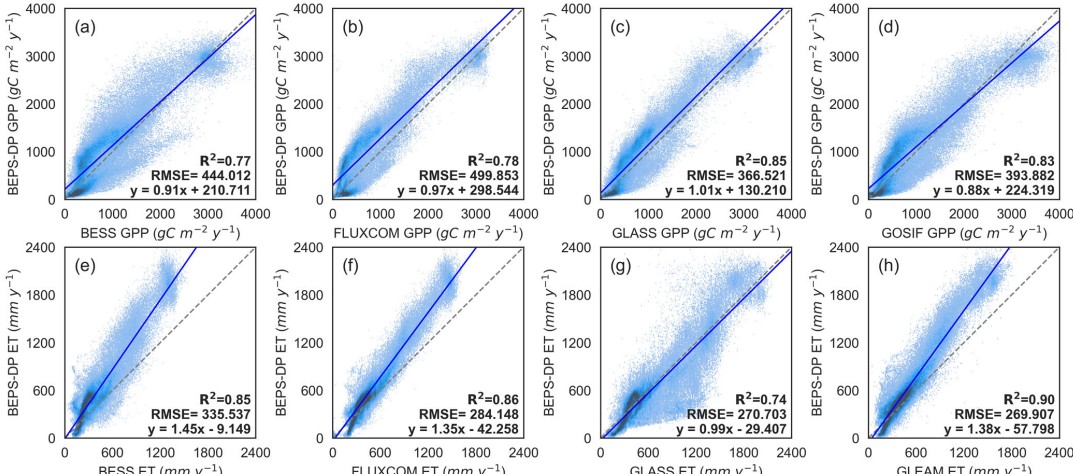

**Figure 8. Comparisons of pixel-to-pixel long-term (2001-2020) averaged annual GPP and ET between the BEPS model with dynamic parameterizations (BEPS-DP) and other models: (a) – (d) The GPP comparisons between BEPS-DP and BESS (Li et al., 2023), FLUXCOM (Jung et al., 2020), GLASS (Liang et al., 2021), and GOSIF (Li and Xiao, 2019) GPP products, respectively; (e) – (f) The ET comparisons between BEPS-DP and BESS, FLUXCOM, GLASS, and GLEAM (Martens et al., 2017) ET products.**

Despite significant advancements in remote sensing technology, ground observations, and the theoretical modeling of carbon and water fluxes, there persists a considerable level of uncertainty in global and regional estimates of GPP and ET (Zheng et al., 2020; Ryu et al., 2019; Li et al., 2023). To investigate the spatial correlations between BEPS-DP and other models, we compared the pixel-to-pixel averaged annual GPP and ET of all available years of each dataset (Figure 8). The spatial pattern of GPP estimated from BEPS-DP correlates well with all other GPP datasets, with $R^2$ ranging from 0.77 to 0.85. The spatial

distribution of ET obtained from BEPS-DP also exhibits strong correlations with all other ET datasets, with $R^2$ ranging from 0.74 to 0.90. The high spatial correlations observed between GPP and ET derived from the BEPS-DP and other existing datasets underscore the consistency of this new hourly dataset with respect to spatial patterns observed in other datasets. Differing from the close magnitudes of GPP estimates between BEPS-DP and other GPP datasets (slope ranging from 0.88 to 1.01), it shows big discrepancies in the magnitudes of ET estimates between BEPS-DP and other ET datasets (slope ranging

from 0.99 to 1.45). BEPS-DP overall produced larger ET than other ET products. Nevertheless, our confidence is substantiated not only in the spatial pattern but also in the absolute magnitude of ET derived from BEPS-DP. This assurance stems from the training of key parameters using ground flux measurements, and the modeled summed ET values align closely with flux data, exhibiting a disparity of less than 9.6% (Figure 3).



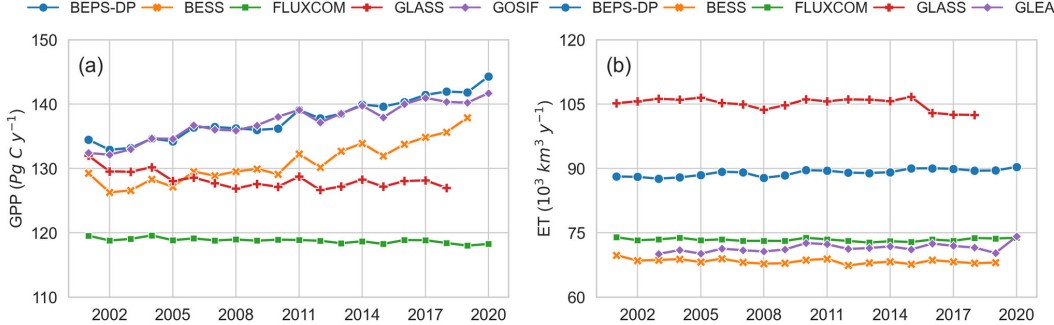


**Figure 9. Comparisons of annual global GPP and ET estimates during 2001-2020: (a) The annual summed GPP between the BEPS model with dynamic parameterizations (BEPS-DP) and BESS, FLUXCOM, GLASS, and GOSIF GPP products; (b) The annual summed ET between BEPS-DP and BESS, FLUXCOM, GLASS, and GLEAM ET products.**

There is substantial discrepancy in the interannual variability and trend of GPP among different datasets while consistent trends of ET with varying magnitudes were observed in different datasets (Figure 9). The interannual variability (standard deviation) of GPP ranges from 0.39 (FLUXCOM), 1.32 (GLASS), 2.81 (GOSIF), 3.13 (BESS), to 3.14 (BEPS-DP) Pg C yr$^{-1}$, with trends varying from -0.005 (FLUXCOM), -0.17 (GLASS), 0.47 (GOSIF), 0.53 (BEPS-DP), to 0.53 (BESS) Pg C yr$^{-2}$. The GPP estimated from machine learning methods, FLUXCOM and GLASS, exhibits no discernible trend during 2001-2020, which

can have resulted from lack of consideration of the $CO_2$ fertilization effect. BEPS-DP exhibits average annual summed GPP that closely align with the GOSIF dataset but demonstrates the highest degree of similarity in terms of interannual variability and trends when compared to BESS. SIF, as the proxy of photosynthetic activity, has been used to quantify GPP at the global scale (Frankenberg et al., 2011; Mohammed et al., 2019) from the plant physiological perspective. BEPS-DP not only provides GPP estimates close to the GPP inferred from SIF but also possesses the capability to elucidate the underlying mechanisms

involved in the carbon cycles as a process-based model.

The interannual variability (standard deviation) of ET ranges from 0.35 (FLUXCOM), 0.54 (BESS), 0.80 (BEPS-DP), 0.98 (GLEAM), to 1.30 (GLASS) × 10$^3$ km$^3$ yr$^{-1}$, with trends varying from -0.12 (GLASS), -0.05 (FLUXCOM), -0.05 (BESS), 0.10 (BEPS-DP), to 0.10 (GLEAM) × 10$^3$ km$^3$ yr$^{-2}$. BEPS-DP demonstrates a close correspondence in terms of not only the average annual summed ET but also the interannual variability and trends with the GLEAM dataset, although a discrepancy

in the magnitude of ET exists between the two datasets. GLEAM estimates global ET by employing data assimilation techniques that integrate remote sensing data with ground measurements (Martens et al., 2017; Miralles et al., 2011) while BEPS-DP incorporates plant physiological processes on the basis of remote sensing and measured fluxes. Incorporating ground measured data contributes to comparable interannual variability and trends in ET estimates between GLEAM and BEPS-DP. However, accounting for the photosynthesis process can lead to improved quantification of transpiration within in the ET

estimates (Chen and Liu, 2020), which results in the different magnitudes of ET estimates from GLEAM and BEPS-DP.



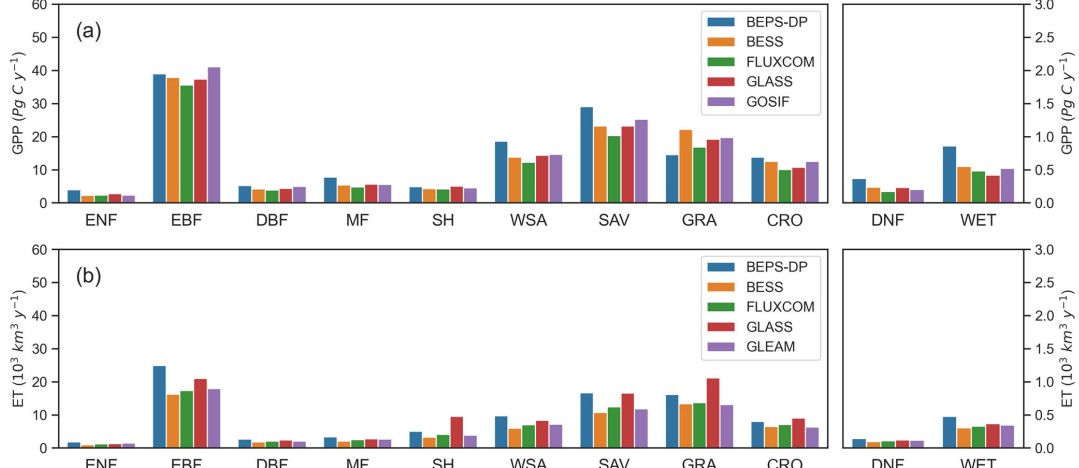

**Figure 10. Comparisons of long-term (2001-2020) averaged annual GPP and ET between the BEPS model with dynamic parameterizations (BEPS-DP) and other models across plant functional types (PFTs): (a) The GPP comparison between BEPS-DP and BESS, FLUXCOM, GLASS, and GOSIF GPP products; (b) The ET comparison between BEPS-DP and BESS, FLUXCOM, GLASS, and GLEAM ET products. The abbreviations for PFTs are: ENF, evergreen needleleaf forests; EBF, evergreen broadleaf forests; DBF, deciduous broadleaf forests; MF, mixed forests; SH, shrublands; WSA, woody savannas; SAV, savannas; GRA, grasslands; CRO, croplands; DNF, deciduous needleleaf forests; WET, wetlands.**

We also compared the long-term averaged annual GPP and ET between BEPS-DP and other datasets across PFTs (Figure 10). The magnitude of GPP estimated from BEPS-DP is generally close to that from other products in forest ecosystems, while small discrepancies exhibit in savannas and grasslands (Figure 10a). The potential overestimates of GPP in savannas may stem from the scarcity of measured fluxes available for those particular ecosystems. Therefore, BEPS-DP has limited access to the necessary information for accurate parameterization, thus contributing to the challenge of achieving precise estimates in this context. The underestimation of GPP in grassland ecosystems may be attributed to the lack of adequately accounting for the C4 plants in BEPS-DP. BEPS-DP generates ET estimates close the GLASS ET dataset in terms of magnitude but tends to yield higher ET values compared to the FLUXCOM, BESS, and GLEAM datasets. The disparity arises because BEPS-DP optimized the key parameters using the measured fluxes but the FLUXCOM, BESS, and GLEAM datasets tend to underestimate the ET when comparing with ground measured water fluxes.

## 4. Discussion

### 4.1 Advantages of this new dataset

The convergence of unparalleled data sources, enhanced computational capabilities, and recent advancements in statistical modeling and machine learning presents promising prospects for expanding our understanding of the terrestrial ecosystems through data-driven approaches (Reichstein et al., 2019). This confluence of factors offers exciting opportunities to unlock new insights and uncover hidden patterns in terrestrial ecosystem processes. In this study, we proposed a novel model that integrate diagnostic process-based models with dynamic parameterizations using machine learning based on measured carbon and water fluxes, which may be regarded a 'handshake' among remote sensing data, a terrestrial biosphere model, and the eddy covariance flux network (Baldocchi, 2020). By combining the advantages, we aimed to enhance the accuracy and reliability of BEPS-DP in capturing the long-term complex dynamics of global carbon and water fluxes.

The diurnal cycling of plant carbon uptake and water use, as well as their responses to water and heat stresses, offer valuable insights for evaluating ecosystem productivity, agricultural production and management practices, carbon and water cycles,



and their interactions with the climate system (Xiao et al., 2021). The hourly timescale of estimated carbon and water fluxes from the diagnostic model with dynamic parameterizations in this study can promote further research on the comprehension and monitoring of the extreme climate events, such as the flash drought occurrence (Christian et al., 2021) and heat waves
(Bastos et al., 2020), and can facilitate deep sights into the diurnal ecosystem functionality, such as the diurnal hysteresis between carbon and water fluxes (Lin et al., 2019) and the impact of physiological drought stress on ecosystems (Zhang et al., 2023a). With the emergence of geostationary satellites and other satellites with high temporal resolutions, this hourly dataset can help elucidate how terrestrial ecosystems respond to diurnal environment conditions in the context of climate change (Xiao et al., 2021; Yamamoto et al., 2023; Jeong et al., 2023).

To make the full advance of the two-leaf BEPS model, the hourly GPP and ET dataset produced in this study also includes hourly GPP and ET from sunlit and shaded leaves separately. These GPP and ET components would also be useful for investigating their distinct responses to meteorological conditions and the coupling between carbon and water fluxes over the diurnal cycle, among many possible uses of this unprecedentedly detailed dataset.

### 4.2 Uncertainties and limitation

Although BEPS-DP can effectively simulate the hourly carbon and water fluxes for most areas of the globe, it may be subject to uncertainties resulting from the lack of representation of eddy covariance fluxes for savannas and tropical forest ecosystems. The allocation of sites is uneven between the northern and southern hemispheres, characterized by a larger number of sites situated in the northern hemisphere (Baldocchi et al., 2001; Baldocchi, 2020), which may also induce some uncertainties in BEPS-DP. The key parameters in stomatal conductance and photosynthesis models, $m$ and $V_{cmax}^{25}$, were optimized from eddy
covariance fluxes in BEPS-DP. Then the optimized $m$ and $V_{cmax}^{25}$ in the ~1 km footprint (Chu et al., 2021) were upscaled to the pixel level during 2001-2020. However, it should be noted that in areas with a limited number of sites, the pixel-level values of $m$ and $V_{cmax}^{25}$ may not sufficiently capture the vegetation physiological status. For example, only one DNF site was included in this study. This limitation arises from the fact that the machine learning algorithm did not incorporate knowledge specifically for such situations. Further work should focus on the enhancement of the machine learning algorithm to improve
the reliability of optimized $m$ and $V_{cmax}^{25}$ particularly in regions with scarce training data. Furthermore, limited training data for the C4 plants in BEPS-DP may result in uncertainties in quantifying GPP and ET in savannas, woody savannas, and crop ecosystems. Efforts will also be made to improve the scheme for simulating the carbon and water fluxes in the C4 plants in BEPS-DP.

Additionally, although the dataset in this study captures the spatial distributions of global GPP and ET at an hourly scale, this
dataset only has a spatial resolution of 0.25°×0.25° due to the climate forcing data resolution and computation capacity. Due to surface heterogeneity and the nonlinear algorithm in BEPS-DP, the estimated GPP and ET fluxes in 0.25°×0.25° pixels would be biased to some extent even if the simulated values at the site level are unbiased (Chen, 1999; Chen et al., 2013). With the advancement and evolution of computational capacity and techniques, such as cloud computing and supercomputing, future research can refine the spatial resolution of the hourly dataset from the current 0.25° degree to higher resolutions, to eliminate
the scale mismatch between the flux tower footprints and the hourly datasets and thoroughly comprehend the intrinsic processes in global carbon and water cycles (Li et al., 2008; Kong et al., 2022).

### 5. Code and Data Availability

The 0.25° × 0.25° global hourly two-leaf GPP and ET datasets for 2001-2020 are available at https://doi.org/10.5281/zenodo.8240492 (Leng et al., 2023). The datasets are provided in NetCDF4 format. The GPP datasets
include two components, the hourly GPP of sunlit and shaded leaves. The ET datasets include three components, the hourly evapotranspiration, transpiration of sunlit and shaded leaves. Each hourly NetCDF4 file represents the GPP/ET in a year at an





hourly scale (g C m$^{-2}$ h$^{-1}$/mm h$^{-1}$). The accumulated daily GPP and ET datasets for 2001-2020 are also available in the same directory. Each daily NetCDF4 file represents the GPP/ET in a year at a daily scale (g C m$^{-2}$ d$^{-1}$/mm d$^{-1}$).

The code for single-pixel hourly Biosphere-atmosphere Exchange Process Simulator (BEPS) v4.10 can be found at https://github.com/JChen-UToronto/BEPS_hourly_site. The detailed descriptions of each module and the user guide for using the hourly BEPS are also included.

For any questions on the dataset and the BEPS model, please contact Jing M. Chen, jing.chen@utoronto.ca.

## 6. Conclusions

In this study, we produced a long-term global two-leaf GPP and ET dataset at the hourly time step by integrating a diagnostic TBM (i.e., BEPS) with dynamic parameterizations. We optimized the key photosynthetic parameters using the flux observations and upscaled the optimized parameters to the global for large scale simulation. The BEPS model with dynamic parameterizations is able to simulate the diurnal, seasonal, and interannual variations of the GPP and ET fluxes at 0.25° resolution. The new hourly datasets of GPP and ET were comprehensively evaluated against flux observations and other remote sensing and machine learning-based estimates over the various temporal and spatial scales. The new dataset provides us with a unique opportunity to study carbon and water fluxes at sub-daily time scales and advance our understanding of ecosystem functions in response to transient environmental changes.

## Author Contributions

JL and JMC devised the conceptual ideas. JL, WL, MX, BL, and YY provided and processed the data. JL conducted the investigation and performed formal analysis. JMC acquired the funding. JL, WL, and XL developed the methodology. JMC supervised the findings of this work. WL, XL, MX, JL, RW, CR, and YY verified the results. JL created the figures and drafted the original manuscript. JMC, WL, XL, MX, JL, RW, CR, BL, and YY reviewed and commented on the manuscript.

## Acknowledgements

This work used the eddy covariance data acquired and shared by the FLUXNET community, including the following networks: AmeriFlux, AfriFlux, AsiaFlux, CarboAfrica, CarboEuropeIP, CarboItaly, CarboMont, ChinaFlux, Fluxnet-Canada, GreenGrass, ICOS, KoFlux, LBA, NECC, OzFlux-TERN, TCOS-Siberia, and USCCC. The ERA-Interim reanalysis data are provided by ECMWF and processed by LSCE. The FLUXNET eddy covariance data processing and harmonization was carried out by the European Fluxes Database Cluster, AmeriFlux Management Project and Fluxdata project of FLUXNET, with the support of CDIAC and ICOS Ecosystem Thematic Center, and the OzFlux, ChinaFlux and AsiaFlux offices. The authors thank the U.S. Geological Survey (USGS) for providing the MODIS data and thank the National Oceanic and Atmospheric Administration for providing the $CO_2$ concentration records.

JMC acknowledges support of Natural Science and Engineering Council of Canada (RGPIN-2020-05163) for this research.

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
