# Peer review of "Global datasets of hourly carbon and water fluxes simulated using a satellite-based process model with dynamic parameterizations."

_Earth System Science Data, 2023_

## Author Comment (AC1)

*Global datasets of hourly carbon and water fluxes simulated using a satellite-based process model with dynamic parameterizations*

**Responses to the comments on essd-2023-328 by Anonymous Referee #2**

Jiye Leng et al.

*The paragraphs in blue Italic indicate the corresponding revised paragraphs in the manuscript. The under-reviewed papers will be preprinted in the final manuscript before the publication.*

1. Need particular explanation on the newly revised BEPS regarding how exactly the hourly GPP and ET were simulated but not from the previous version. If BEPS was designed to simulate the hourly products, were if just because the hourly inputs were not available before?

Thanks for the question. The newly revised BEPS v4.10 keeps the original structure and algorithms but with new standardized framework using the Doxygen format and Git version control. The name for BEPS was also revised from the former '*Boreal Ecosystem Productivity Simulator*' to '*Biosphere-atmosphere Exchange Process Simulator*'. We also open-sourced the BEPS model after the code standardization. BEPS adopts hourly meteorological variables to simulate carbon and water fluxes using the same algorithms in Chen et al. (2012). Besides, BEPS has been comprehensively evaluated in several site-level, showing its capacity to simulate gross primary productivity (GPP) and evapotranspiration (ET) comparable to the eddy covariance measurements (Gonsamo et al., 2013; Luo et al., 2018; Luo et al., 2019).

Due to the computational capacity and data volume, the global hourly GPP and ET dataset based on BEPS has not yet been published before. In this study, compared to the previous research papers on BEPS, we adopted dynamic parameterizations to improve the accuracy of simulated carbon and water fluxes, and presented and analyzed the global GPP and ET in 2001-2020 at the hourly timescale for the first time.

References:

Chen, J.M., Mo, G., Pisek, J., Liu, J., Deng, F., Ishizawa, M., & Chan, D. (2012). Effects of foliage clumping on the estimation of global terrestrial gross primary productivity. *Global Biogeochemical Cycles*, 26Gonsamo, A., Chen, J.M., Price, D.T., Kurz, W.A., Liu, J., Boisvenue, C., Hember, R.A., Wu, C., & Chang, K.-H. (2013). Improved assessment of gross and net primary productivity of Canada's landmass. *Journal of Geophysical Research: Biogeosciences*, 118, 1546-1560

Luo, X., Chen, J.M., Liu, J., Black, T.A., Croft, H., Staebler, R., He, L., Arain, M.A., Chen, B., Mo, G., Gonsamo, A., & McCaughey, H. (2018). Comparison of Big-Leaf, Two-Big-Leaf, and Two-Leaf Upscaling Schemes for Evapotranspiration Estimation Using Coupled Carbon-Water Modeling. *Journal of Geophysical Research: Biogeosciences*, 123, 207-225

Luo, X., Croft, H., Chen, J.M., He, L., & Keenan, T.F. (2019). Improved estimates of global terrestrial photosynthesis using information on leaf chlorophyll content. *Global Change Biology*, 25, 2499-2514

2. Need a better presentation and validation on the newly optimized key photosynthesis and stomatal conductance model parameters (i.e., $V_{cmax}$ and $m$). Was it only revised for the flux sites (as presented in Figure 2)? How were they interpolated into the global scales and what are their uncertainties? How about the spatial and temporal variations of these parameters globally? Also, what will be the differences in the simulated GPP and ET between the new dynamic parameters and original fixed parameters in terms of accuracy and spatial pattern?

Thanks for the suggestion. As you are concerned, there are several problems that need to be addressed. This manuscript focuses on generating and sharing the reliable GPP and ET dataset. We included the global distributions of $m$ and $V_{cmax}$ in another paper which focuses on discussing the trend and spatial patterns of $m$ and $V_{cmax}$ (Leng et al., under review). However, we would add one chapter to include the spatial distributions, species distributions, and validations of $m$ and $V_{cmax}$ in this dataset manuscript to convince the readers on the accuracy and reliability of the dataset by Leng et al. (under review).

We included the validations of the Random Forest regressor for $m$ and $V_{cmax}$ estimations by courtesy of Leng et al. (under review), as shown in **Error! Reference source not found.**. The Random Forest regressor estimates $m$ and $V_{cmax}$ with good agreements to the optimized $m$ and $V_{cmax}$ from measured fluxes in both the training sets and the validation set. The Random Forest regressor can estimate $m$ with $R^2 = 0.95$ and RMSE = 1.414 in the training set (Figure 1a), and $R^2 = 0.59$ and RMSE = 3.663 in the independent validation set (Figure 1b). The Random Forest regressor can estimate $V_{cmax}$ with $R^2 = 0.98$ and RMSE = 4.191 µmol m$^{-2}$ s$^{-1}$ (Figure 1c), and $R^2 = 0.84$ and RMSE = 10.598 µmol m$^{-2}$ s$^{-1}$ in the independent validation set (Figure 1d). Most of the scatter points locate beside the 1:1 line in both the training and validation set, showing the good accuracy of the Random Forest regressor for $m$ and $V_{cmax}$ estimation. The Random Forest regressors build the bridge that links $m$ and $V_{cmax}$ derived from measured fluxes to the gridded data that can be expanded to global coverage and long timeseries.

[Figure]

Figure 1. Comparisons of estimated $m$ and $V_{cmax25}$ from the Random Forest regressor and optimized $m$ and $V_{cmax25}$ from measured fluxes in the training set (a), (c) and in the independent validation set (b), (d) of the Random Forest Regressor, respectively. The color

indicates the scatter density in each plot and the dotted lines indicate the 1:1 line in each plot. Courtesy of Leng et al. (under review).

The global distributions of retrieved $m$ and $V_{cmax}$ are shown in Figure 2a and Figure 3a, and PFT-dependent patterns of $m$ and $V_{cmax}$ are observed in Figure 2b and Figure 3b. We also included the monthly spatial patterns of global $m$ and $V_{cmax}$ during 2001-2020 in Figure 4 and Figure 5, respectively. Strong seasonal variations in $m$ and $V_{cmax}$ are observed in boreal regions while $m$ and $V_{cmax}$ in subtropical and tropical regions are fairly constant within a year.

[Figure]

*Figure 2. The spatial pattern of global $m$ (a) and the averaged $m$ in each PFT (b) during 2001-2020. Courtesy of Leng et al. (under review).*

[Figure]

*Figure 3. The spatial pattern of global $V_{cmax}$ (a) and the averaged $V_{cmax}$ in each PFT (b) during 2001-2020. Courtesy of Leng et al. (under review).*

[Figure]

*Figure 4. Monthly spatial patterns of global  m  during 2001-2020. (a) – (l) averaged  m  from January to December, respectively. Courtesy of Leng et al. (under review).*

[Figure]

*Figure 5. Monthly spatial patterns of global  m  during 2001-2020. (a) – (l) averaged  m  from January to December, respectively. Courtesy of Leng et al. (under review).*

To further validate the gridded global  $m$  and  $V_{cmax}$ , we compared the global retrievals of  $m$

and $V_{cmax}$ in this study with the $m$ census for various biomes from the review by Miner et al. (2017) and the $V_{cmax}$ field measurements collected from Smith et al. (2019), as shown in Figure 6. $m$ estimates in this study were compared with the mean and standard deviation in Miner et al. (2017) while $V_{cmax}$ observations with the timestamp of measurement were compared with the estimated $V_{cmax}$ in the corresponding time period in 2001-2020. Only 0.25° pixels with more than three $V_{cmax}$ measurements were selected in the comparison. The estimated $m$ in the global retrievals agrees well with the measured $m$, with $R^2 = 0.62$ (P = 0.06) and the estimated $V_{cmax}$ in the global retrievals agrees well with the measured $V_{cmax}$, with $R^2 = 0.59$ (P < 0.001).

[Figure]

*Figure 6. Left: Comparison between the PFT-scale mean values of estimated $m$ from the Random Forest regressor and the measured $m$ values reported in the review by Miner et al. (2017). Each horizontal and vertical bar represents the mean $m \pm 1$ standard deviation in the literature values and the estimated values, respectively. The sample sizes of measured $m$ for each PFT are $n = 23$ (ENF), $n = 23$ (EBF), $n = 54$ (DBF), $n = 11$ (SH), $n = 5$ (GRA), and $n = 53$ (CRO). Right: Comparison between the PFT-scale mean values of predicted $V_{cmax}$ from the Random Forest regressor and the measured $V_{cmax}$ values reported in the review by Smith et al. (2019). Courtesy of Leng et al. (under review).*

References:

Leng, J., Chen, J.M., Li, W., Luo, X., Xu, M., Rogers, C., Yan, Y.: Declining global sensitivity of stomatal conductance to photosynthesis. Submitted to *Global Change Biology* (under review)

Miner, G.L., Bauerle, W.L., & B**Error! Reference source not found.**aldocchi, D.D. (2017). Estimating the sensitivity of stomatal conductance to photosynthesis: a review. Plant, Cell & Environment, 40, 1214-1238

Smith, N.G., Keenan, T.F., Colin Prentice, I., Wang, H., Wright, I.J., Niinemets, U., Crous, K.Y., Domingues, T.F., Guerrieri, R., Yoko Ishida, F., Kattge, J., Kruger, E.L., Maire, V., Rogers, A., Serbin, S.P., Tarvainen, L., Togashi, H.F., Townsend, P.A., Wang, M., Weerasinghe, L.K., & Zhou, S.X. (2019). Global photosynthetic capacity is optimized to the environment. Ecological Letter, 22, 506-517

3.   Figure 1. The bottom-up order is quite counter-intuitive to readers. Suggest using top-down order.

Thanks for the suggestion. We have revised the Figure 7 as the top-down order.

[Figure]

*Figure 7. Schematic overview of the methodology and data products of the BEPS model with dynamic parameterizations (BEPS-DP). The flow diagrams show the methodological steps (left) and the details (right) for the BEPS-DP datasets of global hourly two-leaf carbon and water fluxes. SW (shortwave radiation, W m⁻²), TA (air temperature, °C), RH (relative humidity, %), P (precipitation, mm h⁻¹), WS (wind speed, m s⁻¹), GPP (gross primary productivity, g C m⁻² h⁻¹), LE (latent heat, W m⁻²).*

4.   Figure 4, suggest adding the label of 1 in the slope subplots as a reference of good fitting.

Thanks for the suggestion. We have added the label of 1 in the slope and $R^2$ subplots as a reference of good fitting, as shown in Figure 8.

[Figure]

*Figure 8. Evaluation of modeled hourly and daily fluxes against the eddy covariance data in the independent validation set: site-year percentage of $R^2$ in (a) hourly GPP; (c) hourly ET; (i) daily GPP; (k) daily ET; site-year percentage of regression slopes in (b) hourly GPP; (d) hourly ET; (j) daily GPP; (l) daily ET; the mean and standard deviation (SD) of $R^2$ in each PFT in (e) hourly GPP; (g) hourly ET; (m) daily GPP; (o) daily ET; the mean and standard deviation of regression slopes in each PFT in (f) hourly GPP; (h) hourly ET; (n) daily GPP; (p) daily ET. The grey lines indicate 1.0 in $R^2$ and regression slopes as a reference of good fitting. The dashed grey lines in (e) – (h) and (m) – (p) indicate the mean of $R^2$ and regression slopes for all PFTs in GPP and ET.*

5. Need better presentations on the diurnal patterns of GPP and ET for different vegetation function types. For example, providing hourly curves for different vegetation function types and validated against flux site observations. It is still unclear whether these products can capture the diurnal variations of GPP and ET.

Thanks for your suggestions. Since there are 20% of all the sites (809 site years) in the

independent validation set for the comparisons of modeled and measured GPP and ET, it would be too redundant to show all the diurnal patterns (i.e., the hourly curves) for different vegetation functional types in the manuscript. However, for better presentations of our GPP and ET product simulated based on BEPS with dynamic parameterizations, we randomly selected one site-year per each PFT and showed the simulated GPP and ET against flux site observations in three different stages (i.e., day of year 115-125, 195-205, and 275-285). The sites and site-year we selected in the presentations of diurnal curves were shown in the table below.

| Site Name | IGBP | Year | Lat | Lon |
|-----------|------|------|-----|-----|
| CH-Oe2 | CRO | 2013 | 47.2863 | 7.7343 |
| US-KS2 | SH | 2004 | 28.6086 | -80.6715 |
| DE-Hai | DBF | 2004 | 51.0792 | 10.4530 |
| IT-Cpz | EBF | 2003 | 41.7053 | 12.3761 |
| DE-Tha | ENF | 2004 | 50.9624 | 13.5652 |
| NL-Hor | GRA | 2008 | 52.2404 | 5.0713 |
| JP-SMF | MF | 2005 | 35.2617 | 137.0788 |
| AU-Dry | SAV | 2010 | -15.2588 | 132.3706 |
| CN-Ha2 | WET | 2003 | 37.6086 | 101.3269 |
| US-Ton | WSA | 2011 | 38.4316 | -120.9660 |

The diurnal variations of simulated GPP and ET against flux observations were shown below, in three different stages per each site year (the beginning, peak, and ending of the growing seasons). The $R^2$ for GPP and ET in different stages per each site year were shown on the left-top in each subplot.

[Figure]

[Figure]

[Figure]

[Figure]

Besides, we updated the link of dataset thanks to the National Ecosystem Data Bank for providing the platform for sharing the dataset. The updated links include the hourly two-leaf

GPP and ET dataset (3.3 TB) and the accumulated daily two-leaf GPP and ET dataset (197 GB) in 2001-2020. The corresponding paragraphs (Abstract, Code and Data Availability) in the manuscript was also updated.

*"Abstract*
*... The hourly and accumulated daily GPP and ET estimates are available at*
*https://doi.org/10.57760/sciencedb.ecodb.00163 (Leng et al., 2023a) and*
*https://doi.org/10.57760/sciencedb.ecodb.00165 (Leng et al., 2023b)."*

*"Code and Data Availability*
*The 0.25°× 0.25° global hourly two-leaf GPP and ET datasets for 2001-2020 are available at https://doi.org/10.57760/sciencedb.ecodb.00163 (Leng et al., 2023a). The datasets are provided in NetCDF4 format. The GPP datasets include two components, the hourly GPP of sunlit and shaded leaves. The ET datasets include three components, the hourly evapotranspiration, transpiration of sunlit and shaded leaves. Each hourly NetCDF4 file represents the GPP/ET in a year at an hourly scale ($g\ C\ m^{-2}\ h^{-1}/mm\ h^{-1}$). The accumulated daily GPP and ET datasets for 2001-2020 are available at https://doi.org/10.57760/sciencedb.ecodb.00165 (Leng et al., 2023b). Each daily NetCDF4 file represents the GPP/ET in a year at a daily scale ($g\ C\ m^{-2}\ d^{-1}/mm\ d^{-1}$) ..."*

References:
Leng, J., Chen, J.M., Li, W., Luo, X., Xu, M., Liu, J., Wang, R., Rogers, C., Li, B., Yan, Y., 2023a. Global Datasets of Hourly Carbon and Water Fluxes Simulated Using a Satellite-based Process Model with Dynamic Parameterizations [DS/OL]. V1. Science Data Bank.
https://doi.org/10.57760/sciencedb.ecodb.00165. DOI: 10.57760/sciencedb.ecodb.00165.
Leng, J., Chen, J.M., Li, W., Luo, X., Xu, M., Liu, J., Wang, R., Rogers, C., Li, B., Yan, Y., 2023b. Global Datasets of Daily Carbon and Water Fluxes Simulated Using a Satellite-based Process Model with Dynamic Parameterizations [DS/OL]. V1. Science Data Bank.
https://doi.org/10.57760/sciencedb.ecodb.00165. DOI: 10.57760/sciencedb.ecodb.00165.

---

## Author Comment (AC2)

*Global datasets of hourly carbon and water fluxes simulated using a satellite-based process model with dynamic parameterizations*

**Responses to the comments on essd-2023-328 by Anonymous Referee #1**

Jiye Leng et al.

*The paragraphs in blue Italic indicate the corresponding revised paragraphs in the manuscript. The under-reviewed papers will be preprinted in the final manuscript before the publication.*

1. Vegetation transpiration is simulated by the supplementary Eqn. 8. Did the authors set the same Rn-G to shaded and sunlit leaf? Obviously, there are large differences of latent heat for shaded and sunlit leaves.

Thanks for the questions. In BEPS (Biosphere-atmosphere Ecosystem Productivity Simulator), the whole canopy was separated into four groups based on the location and the radiation characteristics of the leaves, including the sunlit leaves in the overstory, the shaded leaves in the overstory, the sunlit leaves in the overstory, and the shaded leaves in the understory (Chen et al., 1999; Liu et al., 2003). The net radiation ($R_n$) on a leaf comprises three sources of radiation:

$$R_n = R_{dir} + R_{diff} + R_L$$

where $R_{dir}$, $R_{diff}$, and $R_L$ refer to the net direct incoming solar radiation, net diffuse solar radiation, and net longwave radiation on the leaf, respectively. For a shaded leaf, $R_{dir} = 0$.
A semi-empirical equation is applied to separate the incoming solar radiation into direct and diffuse parts, expressed as:

$$\frac{S_{dif}}{S_{irr}} = \begin{cases} 0.943 + 0.734r - 4.9r^2 + 1.796r^3 + 2.058r^4, r < 0.8 \\ 0.13, r \geq 0.8 \end{cases}$$

$$S_{dir} = S_{irr} - S_{dif}$$

where $S_{irr}$, $S_{dif}$, and $S_{dir}$ refer to the incident solar irradiance, incoming diffuse solar radiation, and incoming direct solar radiation, respectively. $r$ is indicating the cloudiness of the sky:

$$r = \frac{S_{irr}}{S_0 cos\theta}$$

where $S_0$ is the solar constant as 1362 W m$^{-2}$ and $\theta$ is the solar zenith angle.
Then the net direct solar radiation on the sunlit overstory or understory leaves is calculated as:

$$R_{dir\_o\_sunlit} = R_{dir\_u\_sunlit} = (1 - \alpha_L)S_{dir}cos\alpha/cos\theta$$

where $\alpha_L$ is the albedo of leaves, $\alpha$ is the mean angle between leaf and sun which is set as 60° assuming the canopy has a spherical leaf distribution.
The net diffuse solar radiation of four individual leaf groups are calculated as:

$$R_{dif\_o\_sunlit} = R_{dif\_o\_shaded} = (1 - \alpha_L)(S_{dif}\frac{1 - e^{-0.5\Omega LAI_o/\overline{cos\theta_o}}}{LAI_o} + C_o)$$

$$R_{dif\_u\_sunlit} = R_{dif\_u\_shaded} = (1 - \alpha_L)\left(S_{dif}\, e^{-0.5\Omega LAI_o/\cos\overline{\theta_o}}\, \frac{1 - e^{-0.5\Omega LAI_u/\cos\overline{\theta_u}}}{LAI_u} + C_u\right)$$

where $\Omega$ is the clumping index, $LAI_o$ and $LAI_u$ refer to overstory LAI and understory LAI. $C_o$ and $C_u$ are parameters that quantify the multiple scattering of the direct solar radiation (Chen et al., 1999):

$$C_o = 0.07\Omega S_{dir}(1.1 - 0.1LAI_o)e^{-\cos\theta}$$
$$C_u = 0.07\Omega S_{dir}e^{-0.5\Omega LAI_o/\cos\theta}(1.1 - 0.1LAI_u)e^{-\cos\theta}$$

$\overline{\theta}$ is the representative zenith angle for diffuse radiation transmission calculated as (Liu et al., 2003):

$$\cos\overline{\theta} = 0.537 + 0.025LAI$$

for sunlit and shade leaves.

The net longwave radiation reaching the four leave groups is calculated as:

$$R_{L\_o\_sunlit} = R_{L\_o\_shaded}$$
$$= \frac{1}{LAI_o}\left\{\left\{\varepsilon_o\left[\varepsilon_\alpha \sigma T_\alpha^4 + \varepsilon_u T_u^4\left(1 - e^{-0.5LAI_u\Omega/\cos\overline{\theta_u}}\right) + \varepsilon_g \sigma T_g^4 e^{-0.5LAI_u\Omega/\cos\overline{\theta_u}}\right]\right.\right.$$
$$\left. - 2\varepsilon_o\sigma T_o^4\right\}\left(1 - e^{-0.5LAI_o\Omega/\cos\overline{\theta_o}}\right)$$
$$+ \varepsilon_o(1 - \varepsilon_u)\left(1 - e^{-0.5LAI_u\Omega/\cos\overline{\theta_u}}\right)\left[\varepsilon_\alpha \sigma T_\alpha^4 e^{-0.5LAI_o\Omega/\cos\overline{\theta_o}}\right.$$
$$\left.\left. + \varepsilon_o\sigma T_o^4\left(1 - e^{-0.5LAI_o\Omega/\cos\overline{\theta_o}}\right)\right]\right\}$$

$$R_{L\_u\_sunlit} = R_{L\_u\_shaded}$$
$$= \frac{1}{LAI_u}\left\{\left\{\varepsilon_u\left[\varepsilon_\alpha \sigma T_\alpha^4 e^{-0.5LAI_o\Omega/\cos\overline{\theta_o}} + \varepsilon_o\sigma T_o^4\left(1 - e^{-0.5LAI_o\Omega/\cos\overline{\theta_o}}\right)\right.\right.\right.$$
$$\left.\left. + \varepsilon_g\sigma T_g^4\right] - 2\varepsilon_u\sigma T_u^4\right\}\left(1 - e^{-0.5LAI_u\Omega/\cos\overline{\theta_u}}\right)$$
$$+ \varepsilon_u(1$$
$$- \varepsilon_g)\left\{\left[\varepsilon_\alpha \sigma T_\alpha^4 e^{-0.5LAI_o\Omega/\cos\overline{\theta_o}}\right.\right.$$
$$\left.\left. + \varepsilon_o\sigma T_o^4\left(1 - e^{-0.5LAI_o\Omega/\cos\overline{\theta_o}}\right)\right]e^{-0.5LAI_u\Omega/\cos\overline{\theta_u}}\right.$$
$$\left. + \varepsilon_u\sigma T_u^4\left(1 - e^{-0.5LAI_u\Omega/\cos\overline{\theta_u}}\right)\right\}$$
$$+ \varepsilon_u(1 - \varepsilon_o)\left[\varepsilon_u\sigma T_u^4\left(1 - e^{-0.5LAI_u\Omega/\cos\overline{\theta_u}}\right) + \varepsilon_g\sigma T_g^4 e^{-0.5LAI_u\Omega/\cos\overline{\theta_u}}\right](1$$
$$\left. - e^{-0.5LAI_o\Omega/\cos\overline{\theta_o}}\right)\right\}$$

where $\sigma$ is the Stephen-Boltzmann constant as $5.67 \times 10^{-8}$ W m$^{-2}$ K$^{-4}$. $\varepsilon_\alpha$, $\varepsilon_o$, $\varepsilon_u$, and $\varepsilon_g$ are the emissivity of the atmosphere, overstory, understory, and ground surface, respectively. Therefore, the transpiration in BEPS is simulated separately the sunlit and shaded leaves, considering the different net radiation reaching the sunlit and shaded leave groups. The $R_n -$

$G$ in the Penman-Monteith equation will be different for sunlit and shaded leave groups in the simulation process.

References:

Chen, J.M., Liu, J., Cihlar, J., & Goulden, M.L. (1999). Daily canopy photosynthesis model through temporal and spatial scaling for remote sensing applications. *Ecological Modelling, 124*, 99-119

Liu, J., Chen, J.M., & Cihlar, J. (2003). Mapping evapotranspiration based on remote sensing: An application to Canada's landmass. *Water Resources Research, 39*

2. Line 115: why did you use GLOBMAP LAI data? The spatial resolution of GLOBMAP is quite coarse.

Thanks for the question. The aim of this study is to generate reliable hourly GPP and ET estimates from 2001-2020 at a spatial resolution of 0.25°. Although the GLOBMAP LAI only has a spatial resolution of 0.0727°, the timeseries and the trend of LAI are more important than the spatial resolution regarding to the upscaling from the original spatial resolution to 0.25°.

We used the GLOBMAP LAI data for the following two reasons: 1) The GLOBMAP LAI dataset considered the three-dimensional canopy, which was characterized by the clumping index in the retrieval algorithms (Chen, 2017). For accurate estimations of sunlit and shaded components of GPP and ET, LAI and clumping index are needed, in which the LAI datasets considering the clumping effect is essential (Chen et al., 2012). 2) The GLOBMAP LAI in 2001-2020 used in this study was derived from the MOD09A1C6 land surface reflectance product and the associated illumination and view angles based on the GLOBCARBON LAI algorithm (Chen et al., 2019; Deng et al., 2006; Liu et al., 2012). The algorithm based on the 4-Scale geometric optical model (Chen and Leblanc, 1997) explicitly considered the bidirectional reflectance distribution function on reflectance over the canopy, which was the signals measured by the sensor onboard satellites (Deng et al., 2006). BEPS and the GLOBMAP LAI adopted the same algorithm for separating sunlit and shaded leaves so that it could be reliable to use the GLOBMAP LAI to separate the GPP and ET components in sunlit and shaded leaf groups.

Therefore, the spatial distribution and trends of GLOBMAP LAI are reliable to use for generating the global GPP and ET datasets, especially for the sunlit and shaded components of the simulated carbon and water fluxes.

References:

Chen, J. M. & Leblanc, S. G. (1997) A four-scale bidirectional reflectance model based on canopy architecture. *IEEE Transactions on Geoscience and Remote Sensing* 35, 1316–1337.

Chen, J. M. et al. Effects of foliage clumping on global terrestrial gross primary productivity. *Global Biogeochemical Cycles* 26, GB1019 (2012).

Chen, J. M. in Comprehensive Remote Sensing, Vol. 3 (ed. Liang, S.) Pages 53–77, ISBN 9780128032213, https://doi.org/10.1016/B978-0-12-409548-9.10540-82018 (Elsevier, Oxford, 2017).

Chen, J.M., Ju, W., Ciais, P., Viovy, N., Liu, R., Liu, Y., & Lu, X. (2019). Vegetation structural change since 1981 significantly enhanced the terrestrial carbon sink. *Nature Communications*, 10, 4259

Deng, F., Chen, J. M., Plummer, S., Chen, M. Z. & Pisek, J. (2006). Algorithm for global leaf area index retrieval using satellite imagery. *IEEE Transactions on Geoscience and Remote Sensing* 44, 2219–2229

Liu, Y., Liu, R., & Chen, J.M. (2012). Retrospective retrieval of long-term consistent global leaf area index (1981–2011) from combined AVHRR and MODIS data. *Journal of Geophysical Research: Biogeosciences, 117*

3.  Line 150: it is very important to know the details of optimization algorithm. It seems the parameterization method has not been published. Although the authors mentioned the supplementary materials, but I still did not get how the authors optimize the model parameters. Especially, you mentioned the parameters were optimized for each month at each site-year.

Thanks for the question. The parameters in the coupled photosynthesis-stomata models, $m$ and $V_{cmax}^{25}$, were optimized with an effective global optimization algorithm, the Bayesian Optimization. Bayesian optimization works by constructing a posterior distribution of functions (Gaussian Process) that best describes the simulation to be optimized, described as (Snoek et al., 2012):

| Algorithm: Bayesian Optimization, $X = [m, V_{cmax}]$ |
| --- |
| 1: **for** step = 1, 2, …, **do** |
| 2:     Find $X_{step}$ by optimizing the acquisition function (Gaussian Process): $X_{step} = argmin_X u(X|D_{1:step-1})$. |
| 3:     Update the objective function: $y_{step} = f(X_{step}) + \epsilon_{step}$. |
| 4:     Augment data: $D_{1:step} = \{D_{1:step-1}, (X_{step}, y_{step})\}$, update Gaussian Process. |
| 5: **for end** |

where $argmin$ is an operation that finds the argument that gives the minimum value from a target function, $u$ is the conditional probability, and $\epsilon_{step}$ is the noise following a normal distribution.

The objective function, $f(X)$, to be optimized in this study is cost functions between the simulated variables in BEPS and the measured target values. The Bayesian framework requires a likelihood function combining model and observational errors. In this study, the observation error is regarded as Gaussian white noise, so the likelihood function is treated as the cost function that captures the errors of model simulations. By minimizing the cost functions, the accuracy of the model is optimized and the optimal parameters, $m$ and $V_{cmax}^{25}$, can be estimated.

The best objective function tested in the under reviewed paper is modified mean squared error (MMSE) between modeled and measured fluxes, expressed as (Leng et al., under review):

$$f(X) = |1 - a_{GPP_{obs}}^{GPP_{mod}}| \times \frac{1}{N} \sum \left( \frac{GPP_{mod} - GPP_{obs}}{GPP_{obs}} \right)^2 +$$

$$|1 - a_{ET_{obs}}^{ET_{mod}}| \times \frac{1}{N} \sum \left( \frac{ET_{mod} - ET_{obs}}{ET_{obs}} \right)^2$$

where $N$ is the total number of hourly simulations in a parameter optimization interval, $a_{GPP_{obs}}^{GPP_{mod}}$ is the ordinal least square regression (OLS) slope between $GPP_{mod}$ and $GPP_{obs}$, $a_{ET_{obs}}^{ET_{mod}}$ is the OLS slope between $ET_{mod}$ and $ET_{obs}$. The MMSE in cost functions combines the mean squared error and the deviation of OLS slopes between modeled and observed variables from the identity line. By minimizing MMSE, modeled variables and observed variables can better fit the identity line, which yields the two optimal parameters, $m$ and $V_{cmax}^{25}$, that can generate a model of higher accuracy.

In each round of optimizations, BEPS was run for 500 times to get the $m$ and $V_{cmax}^{25}$ of each month at each site. The monthly approximation is a tradeoff between the stability of the Bayesian parameter optimization and the temporal resolution of the two parameters. Therefore, after the calculation, we can obtain the monthly $m$ and $V_{cmax}^{25}$ for all available sites in the FLUXNET2015 dataset.

References:

Snoek, J., Larochelle, H., & Adams, R.P. (2012). Practical Bayesian Optimization of Machine Learning Algorithms. In, *Advances in Neural Information Processing Systems*: Curran Associates, Inc.

Leng, J., Chen, J.M., Li, W., Luo, X., Rogers, C., Croft, H., Xie, X., Staebler, R.M.: Optimizing seasonally variable photosynthetic parameters based on joint carbon and water flux constraints. Submitted to *Agricultural and Forest Meteorology* (under review)

4. From 2.3.1 and 2.3.2, I assumed the authors first inversed two model parameters m and Vcmax at eddy covariance sites, and then used machine-learning method to generate global gridded dataset of m and Vcmax, and finally, BEPS model was run based on the gridded m and Vcmax to estimate global GPP? if it is so, why the authors did not just upscale GPP from towers to global scale, just like Jung et al. 2009.

Thanks for the question. We first inversed monthly $m$ and $V_{cmax}^{25}$ based on the measured carbon and water fluxes, then upscaled the gridded $m$ and $V_{cmax}^{25}$ using the Random Forest Regressors, and then ran the BEPS model again using the optimized $m$ and $V_{cmax}^{25}$. There are two reasons to explain why we chose this scheme to generate the hourly GPP and ET dataset.

First, the available measured fluxes and meteorological conditions in the FLUXNET2015 dataset only includes data before 2014. If using machine learning method to train the model with measured fluxes and meteorological conditions, there could be biases in data of the extrapolated years after 2015. Besides, the statistical models cannot consider the $CO_2$

fertilization effect in the simulation of carbon and water fluxes, inducing no trends in the long-term GPP and ET (Jung et al., 2019; Jung et al., 2020; Liang et al., 2021), as shown in Figure 9 in the manuscript.

Second, BEPS follows the two-leaf enzyme kinetic scheme, which can simulate reliable GPP and ET with appropriate parameters. Only with meteorological conditions, LAI, and basic parameters, BEPS can generate estimates of GPP and ET per hour in any years, which is not limited by the time range of the measured flux data. The process model also possesses the capability to simulate the details of terrestrial ecosystems, such as the leaf energy balance, stomatal conductance (Chen et al., 1999; Liu et al., 2003; Luo et al., 2018), which cannot be explained by the machine learning upscaling methods. Assimilating the FLUXNET, satellite, and leaf traits (i.e., LAI) data into the process-based models can lead to improved GPP and ET estimates (Ryu et al., 2019). Therefore, the BEPS with dynamic parameterizations can provide reliable estimates of hourly carbon and water fluxes in both sunlit and shaded leaf groups.

References:

Chen, J.M., Liu, J., Cihlar, J., & Goulden, M.L. (1999). Daily canopy photosynthesis model through temporal and spatial scaling for remote sensing applications. *Ecological Modelling, 124*, 99-119

Jung, M., Koirala, S., Weber, U., Ichii, K., Gans, F., Camps-Valls, G., Papale, D., Schwalm, C., Tramontana, G., & Reichstein, M. (2019). The FLUXCOM ensemble of global land-atmosphere energy fluxes. *Scientific Data, 6*, 74

Jung, M., et al. (2020). Scaling carbon fluxes from eddy covariance sites to globe: synthesis and evaluation of the FLUXCOM approach. *Biogeosciences, 17*, 1343-1365

Liang, S., Cheng, J., Jia, K., Jiang, B., Liu, Q., Xiao, Z., Yao, Y., Yuan, W., Zhang, X., Zhao, X., & Zhou, J. (2021). The Global Land Surface Satellite (GLASS) Product Suite. *Bulletin of the American Meteorological Society, 102*, E323-E337

Liu, J., Chen, J.M., & Cihlar, J. (2003). Mapping evapotranspiration based on remote sensing: An application to Canada's landmass. *Water Resources Research, 39*

Luo, X., Chen, J.M., Liu, J., Black, T.A., Croft, H., Staebler, R., He, L., Arain, M.A., Chen, B., Mo, G., Gonsamo, A., & McCaughey, H. (2018). Comparison of Big-Leaf, Two-Big-Leaf, and Two-Leaf Upscaling Schemes for Evapotranspiration Estimation Using Coupled Carbon-Water Modeling. *Journal of Geophysical Research: Biogeosciences, 123*, 207-225

Ryu, Y., Berry, J.A., & Baldocchi, D.D. (2019). What is global photosynthesis? History, uncertainties and opportunities. *Remote Sensing of Environment, 223*, 95-114

5.  ==Besides, the authors did not show the global patterns of m and Vcmax at all, and we cannot judge if their distributions are reliable.==

Thanks for the suggestion. We feel sorry that there are several problems that need to be addressed in the original manuscript. This manuscript focuses on generating and sharing the reliable GPP and ET dataset. We included the global distributions of $m$ and $V_{cmax}$ in another paper which focuses on discussing the trend and spatial patterns of $m$ and $V_{cmax}$ (Leng et al., under review). However, we would add one chapter to include the spatial distributions, species

distributions, and validations of $m$ and $V_{cmax}$ in this dataset manuscript to convince the readers on the accuracy and reliability of the dataset by Leng et al. (under review).

The global distributions of retrieved $m$ and $V_{cmax}$ are shown in Figure 1a and Figure 2a, and PFT-dependent patterns of $m$ and $V_{cmax}$ are observed in Figure 1b and Figure 2b. We also included the monthly spatial patterns of global $m$ and $V_{cmax}$ during 2001-2020 in *Figure 3* and *Figure 4*, respectively. Strong seasonal variations in $m$ and $V_{cmax}$ are observed in boreal regions while $m$ and $V_{cmax}$ in subtropical and tropical regions are fairly constant within a year.

[Figure]

*Figure 1. The spatial pattern of global $m$ (a) and the averaged $m$ in each PFT (b) during 2001-2020. Courtesy of Leng et al. (under review).*

[Figure]

*Figure 2. The spatial pattern of global $V_{cmax}$ (a) and the averaged $V_{cmax}$ in each PFT (b) during 2001-2020. Courtesy of Leng et al. (under review).*

[Figure]

*Figure 3. Monthly spatial patterns of global m during 2001-2020. (a) – (l) averaged m from January to December, respectively. Courtesy of Leng et al. (under review).*

[Figure]

*Figure 4. Monthly spatial patterns of global m during 2001-2020. (a) – (l) averaged m from January to December, respectively. Courtesy of Leng et al. (under review).*

To further validate the gridded global $m$ and $V_{cmax}$, we compared the global retrievals of $m$

and $V_{cmax}$ in this study with the $m$ census for various biomes from the review by Miner et al. (2017) and the $V_{cmax}$ field measurements collected from Smith et al. (2019), as shown in Figure 5. $m$ estimates in this study were compared with the mean and standard deviation in Miner et al. (2017) while $V_{cmax}$ observations with the timestamp of measurement were compared with the estimated $V_{cmax}$ in the corresponding time period in 2001-2020. Only 0.25° pixels with more than three $V_{cmax}$ measurements were selected in the comparison. The estimated $m$ in the global retrievals agrees well with the measured $m$, with $R^2 = 0.62$ (P = 0.06) and the estimated $V_{cmax}$ in the global retrievals agrees well with the measured $V_{cmax}$, with $R^2 = 0.59$ (P < 0.001).

[Figure]

*Figure 5. Left: Comparison between the PFT-scale mean values of estimated $m$ from the Random Forest regressor and the measured $m$ values reported in the review by Miner et al. (2017). Each horizontal and vertical bar represents the mean $m \pm 1$ standard deviation in the literature values and the estimated values, respectively. The sample sizes of measured $m$ for each PFT are $n = 23$ (ENF), $n = 23$ (EBF), $n = 54$ (DBF), $n = 11$ (SH), $n = 5$ (GRA), and $n = 53$ (CRO). Right: Comparison between the PFT-scale mean values of predicted $V_{cmax}$ from the Random Forest regressor and the measured $V_{cmax}$ values reported in the review by Smith et al. (2019). Courtesy of Leng et al. (under review).*

References:

Leng, J., Chen, J.M., Li, W., Luo, X., Xu, M., Rogers, C., Yan, Y.: Declining global sensitivity of stomatal conductance to photosynthesis. Submitted to *Global Change Biology* (under review)

Miner, G.L., Bauerle, W.L., & B4aldocchi, D.D. (2017). Estimating the sensitivity of stomatal conductance to photosynthesis: a review. Plant, Cell & Environment, 40, 1214-1238

Smith, N.G., Keenan, T.F., Colin Prentice, I., Wang, H., Wright, I.J., Niinemets, U., Crous, K.Y., Domingues, T.F., Guerrieri, R., Yoko Ishida, F., Kattge, J., Kruger, E.L., Maire, V., Rogers, A., Serbin, S.P., Tarvainen, L., Togashi, H.F., Townsend, P.A., Wang, M., Weerasinghe, L.K., & Zhou, S.X. (2019). Global photosynthetic capacity is optimized to the environment. Ecological Letter, 22, 506-517

6. And it is also necessary to show the performance of machine learning method to simulate m and Vcmax at eddy covariance towers, which is quite important than GPP. In addition, I am curious that if the authors did not use gridded parameters, and just used site-based inversed parameters to simulate global GPP, how is the performance?

Thanks for the suggestion. We included the validations of the Random Forest regressor for $m$ and $V_{cmax}$ estimations by courtesy of Leng et al. (under review), as shown in Figure 6. The Random Forest regressor estimates $m$ and $V_{cmax}$ with good agreements to the optimized $m$ and $V_{cmax}$ from measured fluxes in both the training sets and the validation set. The Random Forest regressor can estimate $m$ with $R^2$ = 0.95 and RMSE = 1.414 in the training set (Figure 6a), and $R^2$ = 0.59 and RMSE = 3.663 in the independent validation set (Figure 6b). The Random Forest regressor can estimate $V_{cmax}$ with $R^2$ = 0.98 and RMSE = 4.191 µmol m$^{-2}$ s$^{-1}$ (Figure 6c), and $R^2$ = 0.84 and RMSE = 10.598 µmol m$^{-2}$ s$^{-1}$ in the independent validation set (Figure 6d). Most of the scatter points locate beside the 1:1 line in both the training and validation set, showing the good accuracy of the Random Forest regressor for $m$ and $V_{cmax}$ estimation. The Random Forest regressors build the bridge that links $m$ and $V_{cmax}$ derived from measured fluxes to the gridded data that can be expanded to global coverage and long timeseries.

[Figure]

Figure 6. Comparisons of estimated $m$ and $V_{cmax25}$ from the Random Forest regressor and optimized $m$ and $V_{cmax25}$ from measured fluxes in the training set (a), (c) and in the independent validation set (b), (d) of the Random Forest Regressor, respectively. The color indicates the scatter density in each plot and the dotted lines indicate the 1:1 line in each plot. Courtesy of Leng et al. (under review).

For the other question, in the global GPP and ET simulations by BEPS, the input data (i.e., hourly meteorological variables, basic geospatial information, model parameterizations) are independent pixel by pixel. Therefore, we could not only use site-based inversed parameters to simulate global GPP. We can only utilize the site-based inversed parameters to simulate the GPP at the sites to evaluate the performance of dynamic parameterizations. As shown in Figure 7, after adopting dynamic parameterizations, the accuracy of GPP and ET estimates by BEPS significantly improves.

[Figure]

Figure 7. Comparisons of modeled hourly fluxes and measured hourly EC fluxes at 136 sites (809 site years). a) GPP modeled with prescribed parameters, b) GPP modeled with optimized parameters, c) ET modeled with prescribed parameters, d) ET modeled with optimized parameters. $R^2$ and RMSE were calculated between modeled hourly fluxes and measured hourly fluxes at all sites. Courtesy of Leng et al. (under review)

References:

Leng, J., Chen, J.M., Li, W., Luo, X., Xu, M., Rogers, C., Yan, Y.: Declining global sensitivity of stomatal conductance to photosynthesis. Submitted to *Global Change Biology* (under review)

7. Fig. 4 showed the better performance of hourly simulations than daily simulations. Is it possible? I am curious how the authors examine the performance of hourly simulations, and if the authors included all simulations of night and daytime together, which will result in a false high correlation. As this study aimed to produce an hourly GPP and ET, if there are large difference of parameters diurnal scale.

Thanks for the question. The higher performance of hourly simulations can be explained with the following two reasons. First, since the hourly simulations track the diurnal patterns of meteorological conditions, such as air temperature and solar radiation, thus possessing large variances than the daily simulations. Second, the parameter optimization was conducted based on the cost function using the hourly fluxes, in which only the measured fluxes with the quality control flag smaller than two were selected. When we examined the performance of hourly simulations, except for applying the quality control flags to hourly fluxes, we also dropped the extremely small values so as to accurately assess the performance of hourly simulations with BEPS-DP. Therefore, it is reasonable to achieve higher $R^2$ in hourly than daily simulations.

Although this study aimed to generate hourly GPP and ET, $m$ and $V_{cmax}^{25}$, controlled by vegetation physiological status, tend to change seasonally and not significantly change diurnally according to the field measurements (Croft et al., 2017; Luo et al., 2018; Miner and Bauerle, 2017; Smith et al., 2019). Therefore, in this study, we optimized $m$ and $V_{cmax}^{25}$ at the monthly timestep.

References:

Croft, H., Chen, J.M., Luo, X., Bartlett, P., Chen, B., & Staebler, R.M. (2017). Leaf chlorophyll content as a proxy for leaf photosynthetic capacity. *Global Change Biology, 23*, 3513-3524

Luo, X., Croft, H., Chen, J.M., Bartlett, P., Staebler, R., & Froelich, N. (2018). Incorporating leaf chlorophyll content into a two-leaf terrestrial biosphere model for estimating carbon and water fluxes at a forest site. *Agricultural and Forest Meteorology, 248*, 156-168

Miner, G.L., & Bauerle, W.L. (2017). Seasonal variability of the parameters of the Ball–Berry model of stomatal conductance in maize (Zea mays L.) and sunflower (Helianthus annuus L.) under well-watered and water-stressed conditions. *Plant, Cell & Environment, 40*, 1874-1886

Smith, N.G., Keenan, T.F., Colin Prentice, I., Wang, H., Wright, I.J., Niinemets, U., Crous, K.Y., Domingues, T.F., Guerrieri, R., Yoko Ishida, F., Kattge, J., Kruger, E.L., Maire, V., Rogers, A., Serbin, S.P., Tarvainen, L., Togashi, H.F., Townsend, P.A., Wang, M., Weerasinghe, L.K., & Zhou, S.X. (2019). Global photosynthetic capacity is optimized to the environment. *Ecological Letter, 22*, 506-517

8. ==Figure 9: the comparison does not make sense as the products used different LAI datasets, and the different trends basically depends on the trend of LAI datasets.==

Thanks for the question. This manuscript mainly focuses on the generation of a reliable, correct, and accurate global hourly two-leaf GPP and ET dataset using the BEPS model with dynamic parameterizations. The product-to-product comparison aims to present the ranges of current GPP and ET dataset, from the comparisons of magnitudes to spatial distributions. Different products were generated based on various algorithms, including machine learning, process-based models, and empirical or semi-empirical relationships. However, we were focusing on the comparison of the current products but not focusing on the discussions on the different LAI datasets which were used in the retrievals of GPP and ET. Besides, the product-to-product comparison was also conducted in other dataset description papers, i.e., Zheng et al. (2020), to show validate the reliability of the product, so we adopted this dataset comparison method in the discussion part in this manuscript.

Although the different trends basically depend on the trend of LAI datasets, the structure of model and the meteorological driving forces also exert impacts on the trends of simulated long-term carbon and water fluxes. If using the correct LAI dataset generated by the algorithm that considers the complex canopy structure (Liu et al., 2012), the reliable process-based model that couples the Farquhar and Ball-Berry model at the leaf level and upscales the leaf-level fluxes to the canopy level (Luo et al., 2018), and the dynamic parametrizations that originates from the global flux network (Leng et al., under review), we could generate the reliable and accurate global GPP and ET product as described in this manuscript.

References:

Leng, J., Chen, J.M., Li, W., Luo, X., Xu, M., Rogers, C., Yan, Y.: Declining global sensitivity of stomatal conductance to photosynthesis. Submitted to *Global Change Biology* (under review)

Liu, Y., Liu, R., & Chen, J.M. (2012). Retrospective retrieval of long-term consistent global leaf area index (1981–2011) from combined AVHRR and MODIS data. Journal of Geophysical Research: Biogeosciences, 117

Luo, X., Chen, J.M., Liu, J., Black, T.A., Croft, H., Staebler, R., He, L., Arain, M.A., Chen, B., Mo, G., Gonsamo, A., & McCaughey, H. (2018). Comparison of Big-Leaf, Two-Big-Leaf, and Two-Leaf Upscaling Schemes for Evapotranspiration Estimation Using Coupled Carbon-Water Modeling. Journal of Geophysical Research:

Biogeosciences, 123, 207-225

Zheng, Y., Shen, R., Wang, Y., Li, X., Liu, S., Liang, S., Chen, J. M., Ju, W., Zhang, L., and Yuan, W. (2020) Improved estimate of global gross primary production for reproducing its long-term variation, 1982–2017, Earth System Science Data, 12, 27252746, 10.5194/essd-12-2725-2020.

**9. Dataset links update**

Besides, we updated the link of dataset, thanks to the National Ecosystem Data Bank for providing the platform for sharing the dataset. The updated links include the hourly two-leaf GPP and ET dataset (3.3 TB) and the accumulated daily two-leaf GPP and ET dataset (197 GB) in 2001-2020. The corresponding paragraphs (Abstract, Code and Data Availability) in the manuscript were also updated.

*Abstract*

*… The hourly and accumulated daily GPP and ET estimates are available at https://doi.org/10.57760/sciencedb.ecodb.00163 (Leng et al., 2023a) and https://doi.org/10.57760/sciencedb.ecodb.00165 (Leng et al., 2023b).*

*Code and Data Availability*

*The $0.25° \times 0.25°$ global hourly two-leaf GPP and ET datasets for 2001-2020 are available at https://doi.org/10.57760/sciencedb.ecodb.00163 (Leng et al., 2023a). The datasets are provided in NetCDF4 format. The GPP datasets include two components, the hourly GPP of sunlit and shaded leaves. The ET datasets include three components, the hourly evapotranspiration, transpiration of sunlit and shaded leaves. Each hourly NetCDF4 file represents the GPP/ET in a year at an hourly scale ($g\ C\ m^{-2}\ h^{-1}/mm\ h^{-1}$). The accumulated daily GPP and ET datasets for 2001-2020 are available at https://doi.org/10.57760/sciencedb.ecodb.00165 (Leng et al., 2023b). Each daily NetCDF4 file represents the GPP/ET in a year at a daily scale ($g\ C\ m^{-2}\ d^{-1}/mm\ d^{-1}$) …*

References:

Leng, J., Chen, J.M., Li, W., Luo, X., Xu, M., Liu, J., Wang, R., Rogers, C., Li, B., Yan, Y., 2023a. Global Datasets of Hourly Carbon and Water Fluxes Simulated Using a Satellite-based Process Model with Dynamic Parameterizations [DS/OL]. V1. Science Data Bank. https://doi.org/10.57760/sciencedb.ecodb.00165. DOI: 10.57760/sciencedb.ecodb.00165.

Leng, J., Chen, J.M., Li, W., Luo, X., Xu, M., Liu, J., Wang, R., Rogers, C., Li, B., Yan, Y., 2023b. Global Datasets of Daily Carbon and Water Fluxes Simulated Using a Satellite-based Process Model with Dynamic Parameterizations [DS/OL]. V1. Science Data Bank. https://doi.org/10.57760/sciencedb.ecodb.00165. DOI: 10.57760/sciencedb.ecodb.00165.